# Elucidating the protein interaction network of one of the largest icosahedral capsids in the virosphere

Hela Safi [ID] [1,4], Alain Schmitt [ID] [1,4], Alwena Tollec [2], Lucid Belmudes [2], Agathe M G Colmant [ID] [1,3], Olivier Poirot [1], Sebastien Santini [1], Matthieu Legendre [ID] [1], Yohann Couté [ID] [2], Hugo Bisio [ID] [1✉] & Chantal Abergel [ID] [1✉]

## Abstract

Giant viruses challenge traditional boundaries of virology with their large particle sizes, complex genomes, and unique replication strategies. Yet, despite its 750 nm diameter and incorporation of dozens of proteins, the mimivirus virion retains an icosahedral symmetry, a trait often associated with smaller viruses. The functional roles and interactions of most proteins composing such complex icosahedral particles remain elusive. Here, we dissect the spatial and functional organization of mimivirus morphogenesis by integrating bioinformatics, genetics, and interactomics. We performed protein clustering using a structure-informed approach, integrating AlphaFold models with sequence information, to classify and functionally annotate the orphan-protein-rich mimivirus proteome. To map the protein–protein interaction network during morphogenesis, we employed endogenous tagging and co-immunoprecipitation coupled to mass spectrometry. This strategy revealed distinct interaction modules associated with the virion membrane, nucleoid, and viral factory compartments. Comparative analyses with other icosahedral and non-icosahedral giant viruses uncovered conserved assembly nodes and virion-shape-specific adaptations. Our findings shed light on the global organization of mimivirus virion biogenesis and highlight the evolutionary plasticity of viral morphogenetic networks within the *Nucleocytoviricota*.

Subject Categories Microbiology, Virology & Host Pathogen Interaction; Proteomics

## Introduction

Giant viruses have reshaped our understanding of the virosphere, expanding its boundaries far beyond classical definitions. Among the most striking features of these viruses are their immense particle sizes and genomic complexity, a phenomenon known as viral gigantism. This trait has evolved at least three times independently within the phylum *Nucleocytoviricota*, which includes Acanthamoeba polyphaga mimivirus (Raoult et al, 2004), Pithovirus sibericum (Legendre et al, 2014), and Pandoravirus salinus (Legendre et al, 2018) as model representatives (Koonin and Yutin, 2019). Interestingly, while *Pithoviridae* and *Pandoraviridae* have evolved non-icosahedral, ovoid forms (Bisio et al, 2023), members of the family *Mimiviridae* present the canonical icosahedral capsid architecture (Parent et al, 2018). Indeed, icosahedral symmetry is ubiquitous among smaller viruses (Zandi et al, 2004), but mimivirus possesses a 450 nm icosahedral capsid surrounded by long glycosylated fibrils that facilitate host interaction and uptake. How this level of complexity emerged remains elusive.

Mimivirus' particle contains an internal compartment enclosed in a lipid membrane referred to as the nucleoid. This nucleoid harbors the viral genome, exceeding 1.2 Mbp, with over 1000 predicted genes, several of them encoding proteins previously thought to be exclusive to cellular life, such as translation-related components (Xiao et al, 2009; Rodrigues et al, 2015; Raoult et al, 2004; Mutsafi et al, 2014). Mimivirus viral entry is mediated by the fusion of the viral membrane with the host phagosome, after opening of the mimivirus capsid at a unique vertex coined the stargate, which breaks the icosahedral symmetry (Zauberman et al, 2008). Mimivirus replicates entirely within the host cytoplasm, forming virus-induced compartments known as viral factories (VFs) with liquid-like properties (Rigou et al, 2025). These are dynamic hubs where genome replication and virion assembly occur in coordination with the recruitment of membranes, likely derived from the host endoplasmic reticulum (Suárez et al, 2013; De Castro et al, 2013). Previous proteomic and genomic analyses have revealed a conserved core proteome among members of the *Mimiviridae*, particularly proteins associated with the nucleoid and capsid structures (Yutin et al, 2013; Schrad et al, 2020). These findings suggest that the virus relies on a conserved functional gene set to drive its replication and morphogenesis.

[1]Centre National de la Recherche Scientifique, Information Génomique & Structurale (IGS), Unité Mixte de Recherche 7256 (Institut de Microbiologie de la Méditerranée, FR3479), IM2B, IOM, Aix-Marseille University, Marseille, France. [2]Univ. Grenoble Alpes, INSERM, CEA, UA13 BGE, CNRS, CEA, UAR2048, Grenoble, France. [3]Unité Des Virus Emergents (UVE: Aix-Marseille University, Universita Di Corsica, IRD 190, Inserm 1207 IRBA), Marseille, France. [4]These authors contributed equally: Hela Safi, Alain Schmitt. ✉E-mail: hugo.bisio@igs.cnrs-mrs.fr; chantal.abergel@igs.cnrs-mrs.fr

Despite extensive genomic and structural characterization of mimivirus, the functional roles of most viral proteins remain poorly understood. In particular, the protein–protein interaction (PPI) networks that underpin virion assembly and intracellular organization are largely uncharacterized. Mapping mimivirus' interactome is therefore essential to decipher the molecular logic underlying virion morphogenesis and to reveal the organizational principles of viral interaction networks across diverse viral taxonomies, shedding light on their evolutionary trajectories toward gigantism.

This study aims to unravel the spatial, chronological, and functional organization of mimivirus' virion morphogenesis by combining structural predictions and functional approaches. We first employed AlphaFold (AF)-based (Jumper et al, 2021; Mirdita et al, 2022) virome-wide structural clustering to explore the conservation and organization of mimivirus' virion proteome. In addition, we mapped the PPI network during infection using selected endogenously tagged viral proteins (mostly centered on virion and VF) and co-immunoprecipitation (co-IP) coupled to mass spectrometry (IP-MS). Comparative analysis across icosahedral and non-icosahedral giant viruses highlights evolutionary patterns in assembly pathways, revealing both conserved interaction hubs and architecture-specific adaptations.

## Results and discussion

### AF-based structural clustering and annotation of mimivirus proteins

Since the discovery of giant viruses, one of the key questions is the function and origin of their ORFan-rich genomes. To gain functional insight into mimivirus' proteome, and its virion morphogenesis module, we applied the same clustering methodology as Nomburg et al (Nomburg et al, 2024) to build orthogroups at the scale of the virosphere (called virome in Nomburg et al (Nomburg et al, 2024)). Structural information being more conserved than sequence, we improved remote similarity detection during the clustering process using AF-predicted models. Using mimivirus as an example, we show that the available structural information has dramatically improved since the AF revolution, similarly to what was described for the human proteome (Porta-Pardo et al, 2022), from less than 7% of predicted proteins with information being available experimentally or through homology from the PDB (Berman, 2000), to more than 70% of them showing AF models with at least an intermediate confidence level (Fig. 1A). To improve the clustering associated to mimivirus taxon, we extended the original Nomburg et al virome dataset (Nomburg et al, 2024) of 4642 genomes with 62 targeted additional genomes (Giant virus database: GVDB) from the *Megaviricetes* and *Pokkesviricetes* (Fig. 1B,C; Table EV1). These new genomes correspond to 53 new species (spread in 8 orders, 12 families, and 2 classes), populating the mimivirus taxonomical neighborhood and reaching more relevant and well-defined orthogroups. We then applied the same clustering pipeline, based on MMseqs2 (Steinegger and Söding, 2017), retrieving 34,146 sequence clusters and Foldseek (Van Kempen et al, 2024), which led to 27,369 protein clusters (Fig. 1D).

We then selected viral complexes available in the PDB database, namely poxvirus and African swine fever virus (ASFV) RNA polymerase complexes, to verify the presence of the poxvirus and ASFV subunits in the same orthogroups as their known homologs in mimivirus. With the preliminary clustering method, similar to Nomburg et al (Nomburg et al, 2024), only 6 out of 15 poxvirus and ASFV RNA polymerase subunits (5 out of 8 most conserved RNA polymerase subunits) were found in the same clusters (Table EV2) as mimivirus homologs. We have therefore implemented a merge step to identify and merge highly connected protein clusters. This involves an additional all-versus-all structural similarity search, based on which clusters are subsequently merged if over 65% of proteins in one cluster have links to at least one member of the other cluster. Applying the clustering merge step (Figs. 1D and EV1), mimivirus homologs were associated with the poxvirus/ASFV cluster for 10 out of 15 subunits (6 out of 8 most conserved RNA polymerase subunits). Moreover, the merge step led to 1205 fewer clusters (Figs. 1D and EV1D) at the virome+GV scale, and reduced the number of mimivirus clusters from 690 to 613. Importantly, to minimize false-positive results, the above clustering strategy focuses on stringent similarity with 70% coverage of both query and target. Thus, this large-scale clustering strategy is limited in that it overlooks partial homology when domains are gained or lost, such as for proteins R209, L208, R472 or R326_R327 from the RNA polymerase (Table EV2).

To verify the consistency between the updated clustering results and the original results obtained by Nomburg et al (Nomburg et al, 2024), we estimated a Normalized Mutual Information (NMI) score using the Python scikit-learn library. The NMI score of 0.972 obtained for all proteins present in both clustering results shows that the structure of the orthogroups is mostly conserved in our implementation of the pipeline, including an extension of the dataset (Fig. 1D; Appendix Fig. S1A,B).

Next, we estimated the ancestry of all mimivirus' proteins by retrieving the taxonomic level of its clusters' Last Common Ancestor (LCA) (Fig. 1E). Our extension of the dataset around mimivirus' taxon provided important updates on the species to subfamily levels (Appendix Fig. S1A,B), mainly due to the addition of Megavirus, Cotonvirus and Tupanvirus genuses (Figs. 1E and EV1D), the *Klosneuviridae* subfamily, and several pacmanvirus genomes in the *Asfuvirales* order. This analysis revealed that the subfamily level exhibits the highest degree of conservation, corresponding to 34% (210 in Fig. 1E) of the predicted proteome. Moreover, mimivirus showed a higher number of shared clusters with *Klosneuvirinae* (at the family level) or *Algavirales* and *Pimascovirales* (at the class level). Furthermore, the phylum conservation level showed shared protein clusters with *Asfuvirales* and *Chitovirales* corresponding to 13% (81 in Fig. 1E) of the clusters of the total mimivirus proteome (613) and 18% (33) of virion proteome clusters (180) (Fig. 1E). Interestingly, more than 13% (82) of the proteome corresponds to mimivirus genus ORFans (Fig. 1E). Annotation of the total mimivirus proteome using PFAM (Protein Families database) homology and AFDB (AlphaFold Database) clusters matching, provided possible annotation for 65% (resp. 63%) of the total proteome (resp. the virion proteome) (Appendix Fig. S1C), and showed high consistency with the conservation taxonomic level (Fig. 1E). Notably, 95% of the virion proteins conserved at the phylum level harbor a PFAM or AFDB annotation, in contrast to only 40% of those conserved at the subfamily level (Fig. 1E). This observation reinforces the fact that giant viruses encode a large diversity of proteins, only conserved at lower taxonomic levels, due to evolutionary divergence and/or to the emergence of new proteins (Legendre et al, 2018).

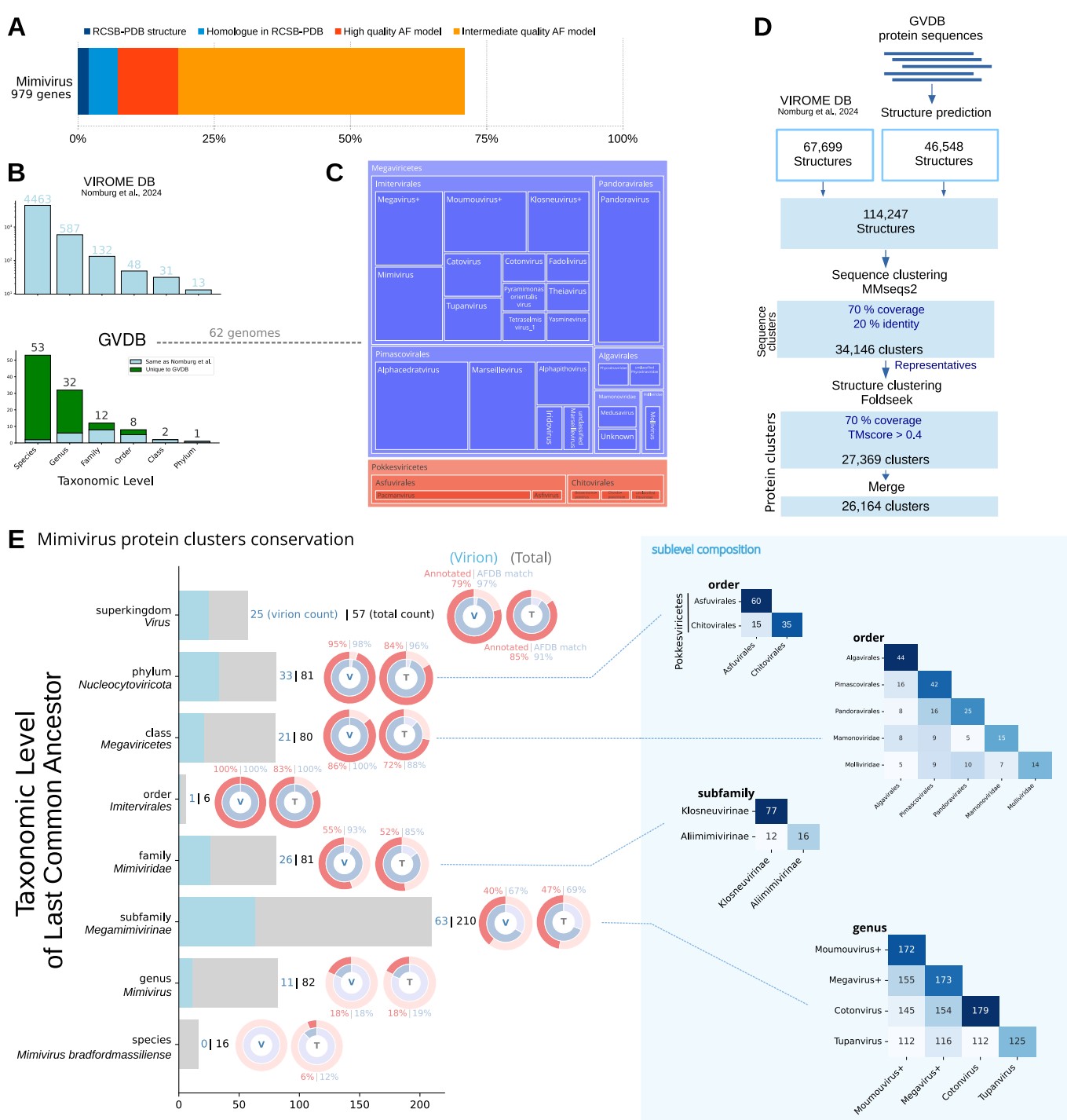

**Figure E. Mimivirus protein clusters conservation**

Importantly, significant improvements have been achieved by using the hybrid structure/sequence clustering method (Fig. EV2) compared to sequence-only clustering. For instance, 39 extra mimivirus protein clusters with homologs at the phylum level were identified (Fig. EV2D). Indeed, sequence clusters from the genus, subfamily or family levels have merged with class, phylum, or superkingdom level clusters while adding structural information (Fig. EV2D). This hybrid sequence+structure strategy also led to higher paralog counts in all bins (Fig. EV2B), and the presence of 3

clusters with at least 21 mimivirus paralogs (21+ bin) corresponding to an Ankirin repeat protein cluster (97 paralogs), a BTBPOZ domain containing cluster (28 paralogs) and a protein cluster of unknown function with several predicted transmembrane domains (22 paralogs). This gain highlighted the ability of structure-informed methods to reveal highly conserved proteins that may contribute to essential roles in viral processes and evolution. To gain a deeper understanding of morphogenesis in icosahedral particles such as mimivirus, we next employed experimental

**Figure 1. The predicted structural proteome of mimivirus.**

(A) AF contribution to mimivirus proteome with High-quality AF model (pLDDT >90) for ~20% of the proteome (Red), intermediate quality model (pLDDT >70) for ~70% of the proteins (Orange). Structural information in the PDB database or inferred by homology with PDB structures accounts for less than 8% of the mimivirus proteome (dark blue and light blue). (B) Virome DB (top) and GVDB (bottom) taxonomic level counts. GVDB is contributing 53 new species, mostly belonging to two classes (*Megaviricetes*, *Pokkesviricetes*). (C) GVDB 62 genomes taxonomic level composition by class, order and genus. (D) Pipeline for protein clustering 114,247 predicted AF models (67699 retrieved from the Nomburg et al (Nomburg et al, 2024; *Virome DB*, 46,548 folded using ColabFold: GVDB). Protein sequences were clustered to 70% coverage and 20% identity. Predicted structures of the 34,146 representatives in each cluster were aligned and clustered together based on 70% coverage and TMscore ≥0.4 (27,369 clusters). A last merge produced 26,164 clusters. (E) Taxonomic Level of the Last Common Ancestor (LCA) of the 613 clusters containing at least one mimivirus member. Left, taxonomic level of the LCA of each mimivirus protein cluster. For example, a protein cluster encoded by viruses from different orders but the same class, are placed in the class row. Clusters are marked in blue or gray when, respectively, present or absent from the virion. The virion proteome was retrieved from Villalta et al (Villalta et al, 2022) as described in "Methods" section. The highest fraction (34% corresponding to 210/613 protein clusters) of mimivirus proteins is conserved at the subfamily level (*Megamimivirinae*). Right, pie chart indicating the fraction of proteins that belong to clusters whose representatives aligned to the AF database (blue), and the fraction of annotated proteins (red), with their representative aligned to a PFAM domain, or an annotated AFDB homolog. Right, sublevel composition matrices. Source data are available online for this figure.

approaches to identify protein interactions and investigate their evolutionary conservation in both icosahedral and non-icosahedral viruses.

## Mapping protein–protein interactions during mimivirus morphogenesis

To gain insight into the spatial and functional organization of the morphogenesis processes occurring during the mimivirus infectious cycle, we endogenously HA-tagged a selection of viral proteins previously identified in virion proteomes (Villalta et al, 2022) (Fig. EV3). To investigate their subcellular localization during virion morphogenesis, we focused on both predicted membrane-associated and soluble proteins. R443, L330, R387, R710, R287, L487, L323, R335, L446, L593, R595, and R513b were predicted to contain at least one transmembrane domain (TMD), based on Phobius, DeepTMHMM, and TMBed ("Methods"). In contrast, R317, L515, L567, L347, L410, L454, L264, R721, and L274, were predicted to be soluble. At 6 h post-infection (hpi) of *Acanthamoeba castellanii*, fluorescence microscopy revealed that the majority of the proteins were localized at the periphery of the VF (Fig. 2). This finding is consistent with previous works (Fridmann-Sirkis et al, 2016; Rigou et al, 2025), which demonstrated through proteomic analysis that several mimivirus proteins are tentatively localized at the VF, where membrane biogenesis and virion assembly occur. Notably, R287 and R317 exhibited distinct localization patterns. The membrane protein R287 showed a punctate cytoplasmic distribution, while R317 appeared to localize within the host cell nucleus (Fig. 2). These observations suggest potential non-structural or regulatory roles and warrant further validation using additional intracellular markers. Control cells infected with wild-type mimivirus showed no specific immunofluorescence signal (Fig. 2). To confirm expression and assess protein size, we performed western blot analysis at 6 hpi for each tagged protein (Appendix Fig. S2).

We then performed co-IP assays for eight of these bait proteins, including five membrane-associated proteins (L330, R443, R387, L323, R595) and three non-membrane proteins (L410, R721, L274) (Fig. 2; Appendix Fig. S2). Co-purified proteins were identified and quantified by MS-based proteomics and compared to negative controls to highlight potential binding partners for each bait protein (Dataset EV1). An interaction network was generated using these co-IP results combined with two co-IP datasets previously

generated with mimivirus proteins associated with the VF, one located in the outer layer (Outer Layer Scaffold 1, OLS1: R561) and the other in the inner layer and nucleoid (Inner Layer Scaffold 1, ILS1: R252) (Rigou et al, 2025). Analysis of shared interactors among the 10 bait proteins revealed two distinct subnetworks within the co-IP interaction map (Fig. 3A,B,D). The first subnetwork, referred to as the Virion-Membrane (VM) subnetwork, is composed predominantly of proteins containing predicted transmembrane domains and exhibits few interactions with host-derived proteins (Fig. 3A, green background). IP-MS of R443, a thioredoxin-domain-containing protein, revealed 15 enriched interactors (14 viral, 1 host) of the bait compared to the negative control, including 9 viral membrane proteins. IP-MS of membrane proteins L323 and R595 as baits, identified 37 and 18 enriched partners, respectively, with substantial overlap: 6 proteins were shared between R443 and L323, and R595 shared 3 with R443 and 15 with L323. Most interactors were membrane-associated; IP-MS of R387 and L330 revealed fewer, primarily host-derived proteins (Dataset EV1; Fig. 3A,B). In contrast, the second subnetwork is associated with nucleoid and VF proteins (NVF subnetwork) and displays extensive interactions with host proteins (Fig. 3A, gray background). Notably, IP-MS of the major core protein L410 revealed 47 interactors, including 29 viral and 18 host proteins. Other bait proteins R252, R561, R721, and L274 exhibited even stronger host associations, with a higher number of host proteins relative to viral ones: 123 vs. 62 for R252, 66 vs. 18 for R561, 39 vs. 26 for R721, and 59 vs. 21 for L274 (Dataset EV1; Fig. 3A). Shared interaction analysis showed that L410 overlapped with R721 and R252 through 34 and 13 common partners, respectively, while R721 and R252 shared 15 interactors (Fig. 3B). These subnetworks, identified by Louvain clustering (Blondel et al, 2008) revealed a consistent organization: nearly all nucleoid and VF proteins were connected within the NVF subnetwork, while most proteins with predicted transmembrane domains were preferentially associated within the VM subnetwork (Fig. 3C). It is interesting to note that the distinct connectivity of the two subnetworks reflects functional specialization. VM proteins primarily interact with other viral proteins, supporting roles in virion assembly and membrane organization as described in previous reports (Kuznetsov et al, 2013; Rodrigues et al, 2021). In contrast, NVF proteins showed extensive host interactions, consistent with their involvement in replication and VF biogenesis, emphasizing their dependence on host machinery. In addition, in mimivirus, the functional

partitioning of the VM and NVF subnetworks mirrors their temporal expression patterns (Fig. EV4) (Legendre et al, 2011; Bessenay et al, 2024). The VM subnetwork, enriched in membrane proteins, mainly corresponded to late gene expression clusters (GV3 to GV5), consistent with its role in virion assembly and morphogenesis. In contrast, the NVF subnetwork, associated not only with morphogenesis but also with replication and VF organization, largely aligns with early expression clusters (GV1 and GV2).

### PPI network: putative connections and biological functions

To infer the biological significance of the PPI network, we performed Gene Ontology (GO) enrichment analysis to identify functional associations likely involved in mimivirus morphogenesis and host interaction (Dataset EV2 Tabs 1-2). GO enrichment was identified for three viral partners of L323, and 55 host partners of 4 additional baits (L274, R252, R561, L330). Host partners of the VM subnetwork bait proteins were found to be enriched in the endoplasmic reticulum for bait L330, and viral partners of L330 were enriched in membrane components. The NVF subnetwork bait proteins were connected to many host proteins with enriched GO terms related to translation for R561 (OLS1) and R252 (ILS1) bait proteins, and oxidation-reduction reactions (Biological Process (BP): cell redox homeostasis, Cellular Component (CC): endoplasmic reticulum, Molecular Function (MF): oxidoreductase activity, acting on a sulfur group of donors) for the L274 bait protein. Interestingly, when using these connected host proteins to search for homologous complexes in the PDB database, several large complexes with functions related to translation were, for instance, identified. Consequently, the human ABCE1-bound 43S pre-initiation complex (7a09), the atypical cytoplasmic ribosome of *Euglena gracilis* (6zj3), and the indirectly connected Human Minor Spliceosome complex (7dvq) were added to the global PPI network (Fig. 3D), providing additional insights for around 40 host proteins connected to the NVF subnetwork. Such subnetworks might be associated with recently identified sub-compartmentalization of the central dogma-associated functions at, or close to, the viral factories (Rigou et al, 2025; Zhang et al, 2018; Mayer et al, 2025).

### Evolutionary insights into protein interaction networks across viral architectures

We next investigated the evolutionary properties of this interaction network by comparing conserved interaction patterns across icosahedral and other virion shapes. For this purpose, we applied the virome+GVDB clustering approach described above, and compared the taxonomic levels of the last common ancestor (LCA) of each mimivirus protein with the topology of the interaction network (Appendix Figs. S3 and EV5).

In the VM and NVF subnetworks, nearly all proteins are at least conserved at the subfamily level, contrary to the rest of the mimivirus proteins (826 proteins with no interaction observed with the baits used in this study), showing a 9.8% genus level ORFan content. This likely reflects a higher conservation of proteins associated to viral particles, but could also indicate that virion proteins are expressed at higher levels than other proteins, making it easier to detect by MS. Globally, the VM and NVF subnetworks display similar evolutionary conservation distributions (Appendix Fig. S3B). It should be noted that horizontal gene transfer (HGT) is thought to occur between distantly related viruses infecting the

same host (Wu et al, 2024), which may affect the accuracy of this LCA analysis.

Along the same line, we classified clusters based on the presence or absence of proteins from viruses with non-icosahedral-shaped virions and used these labels to identify mimivirus proteins specific to icosahedral virions (Fig. EV5). The proportion of icosahedral-related proteins is slightly higher in the VM subnetwork (66%) compared to the NVF subnetwork (51%), especially if only considering proteins that are conserved at least at the taxonomic level corresponding to order (VM: 32%, NVF: 12%) (Fig. EV5B). This trend is recapitulated by analyzing virion proteins where icosahedral-related proteins were more abundant at membrane and other virion fractions, while less represented in the nucleoid proteome (Fig. EV5C). Importantly, such a trend would indicate an independent evolution of the nucleoid relative to the rest of the capsid. However, no statistically significant differences were observed when comparing different icosahedral viruses (Appendix Fig. S4). This suggests that either the available proteomic data and/ or compartment annotations are insufficient to support a robust analysis, or, more likely, that morphological variations among virions are associated with functional adaptations in virion proteins rather than with simple protein composition. This latter hypothesis would be consistent with the evolutionary history of essential proteins involved in maintaining the icosahedral shape, such as the double jelly-roll major capsid protein (DJR-MCP). In mollivirus or poxvirus (non-icosahedral-shaped virions), the DJR-MCP is still used as a scaffold protein to build nascent virions (Hyun et al, 2011). Future studies should focus on dissecting the interaction networks of non-icosahedral viruses to identify potential similarities and differences. Such analyses may help reveal specific networks associated with virion shape rather than with generic functions such as infection.

## Non-icosahedral proteins associated with the membrane subnetwork

Two major hotspots of conservation that breach the otherwise icosahedral conserved pattern were observed within the membrane interaction subnetwork (VM):

### Homologous entry fusion complex

Poxvirus Entry Fusion Complex (EFC) has been previously described (Kao et al, 2023; Diesterbeck et al, 2025) as being composed of 11 proteins with mostly known structures. In the VM subnetwork, we found 6 mimivirus proteins homologously related to 2 poxvirus EFC families, F9/L1 and A16/G9/J5 (Fig. 4A). Interestingly, these proteins are all identified in the IP-MS experiment using L323 as bait. The poxvirus F9/L1 family proteins have been found to be structurally similar to mimivirus L323. Initially, these mimivirus and poxvirus proteins were not found in the same cluster, mostly due to low AF model prediction accuracy (poxvirus F9/VACWR088 mean pLDDT score 55.9) leading to a low TM-score when aligned with mimivirus L323 (TM-score = 0.33) (Fig. 4B). For this specific case, FATCAT flexible structure alignment tool predicted a significant similarity score (*P* val=5e-4, 195 equivalent positions with an RMSD of 2.82 Å and 4 twists), and provided a flexible alignment leading to an improved TM-score of 0.58. Mimivirus L323 cluster and poxvirus F9/VACWR088 cluster have thus been manually merged to reflect this finding, also

pointing to a limitation of the presented clustering strategy based on Foldseek-3Di alignments, that could be improved using a flexible structural alignment step as described in FASSO (Andorf et al, 2022), while keeping in mind that Foldseek is about 40,000 times faster than other approaches (Van Kempen et al, 2024). This manual cluster merge step, driven by flexible alignment, led mimivirus L323 into the non-icosahedral proteins cluster set.

The second poxvirus EFC family A16/G9/J5 has been found to be related to five mimivirus proteins, directly identified by (Kao et al, 2023) (R557) or by structural homology with the tupanvirus deep ocean homologous protein identified by (Kao et al, 2023) (Fig. 4C,D). All four proteins identified through tupanvirus deep ocean proteins show at most partial structural superimposition with poxvirus A16, G9, or J5 (Fig. 4C,D). The N-terminal domains of R486 and L775 or L65 and L778 are not found in poxvirus and could be associated with a new Entry Fusion Complex, specific to *Mimiviridae*. Future efforts will be directed at dissecting this *Mimiviridae* optimized entry fusion complex.

### R443: an example of exaptation of a metabolic enzyme

Previous studies have shown that metabolic enzymes are frequently incorporated into the virion, where they perform structural moonlighting functions (Villalta et al, 2022; Mühlberg et al, 2025; Mutz et al, 2025). R443 is a member of the thioredoxin (Trx) superfamily, which is typically involved in redox pathways through an active site with conserved cysteine residues (Collet and Messens, 2010). Thioredoxins operate through a conserved catalytic mechanism involving the CXXC motif, where the N-terminal cysteine attacks disulfide bonds in target proteins to form a mixed disulfide intermediate, subsequently resolved by the C-terminal cysteine, thereby regenerating thioredoxin and releasing the reduced substrate (Berndt et al, 2008; Buchanan and Balmer, 2005; Arnér, 2009). R443 contained the conserved motif CXXC as shown in Appendix Figs. S5 and S6A. To explore the conservation of R443 homologs across large DNA viruses, we performed a structural alignment of selected proteins within the same cluster. The alignment shows that the thioredoxin-like globular domain is conserved across diverse viral families, supporting a preserved redox function. In contrast, predicted transmembrane domains are only found in members of the *Imitervirales* order, suggesting a lineage-specific membrane association potentially linked to functional specialization (Appendix Fig. S5A). In addition, structural alignment with human thioredoxin (1ERT) confirms that mimivirus R443 retains the conserved thioredoxin fold, including the catalytic CXXC motif (green and red stars) (Appendix Fig. S5B). To further investigate the role of mimivirus R443 protein, we generated an R443 knockout (KO) mutant using homologous recombination, and the clonality of recombinant viruses was demonstrated by genotyping (Fig. 5A). The successful generation of an R443-deficient mimivirus shows that this gene is non-essential for viral replication. Concordantly, either this protein is involved in a fitness-conferring function non-essential for morphogenesis, or functional redundancy could compensate for the absence of the protein, as previously demonstrated for other essential functions in giant viruses (Bisio et al, 2023; Alempic et al, 2024). Indeed, at least two other Trx-like proteins are detected in the virion proteome (Fridmann-Sirkis et al, 2016). To probe whether the Trx activity is the key function of this protein, we generated cis-complemented R443 mutants with wild-type (CXXC) or cysteine-substituted

motifs (SXXC, CXXS) (Fig. 5B), confirmed by genome sequencing (Appendix Fig. S6B). All variants restored localization at the VF with similar expression levels, in contrast to the KO strain (Fig. 5C). Competition assays revealed a fitness cost in the R443 KO across 10 passages, with a phenotypic effect on virus propagation (Fig. 5D). We then compared R443 KO versus the 3 cis-complemented mutants with R443 WT (CXXC) or cysteine-substituted motifs (SXXC, CXXS). Interestingly, the R443 KO fitness loss was similar with the three complemented forms, indicating that the redox-active cysteines were not central to the protein function (Fig. 5D). To further address any potential oxidoreductase activity, we performed IP-MS using CXXS and SXXC mutants and R443 WT as baits, without crosslinkers or reducing agents (Appendix Fig. S6C). Unlike typical thioredoxins, R443 WT failed to retain any specific interacting substrates compared to mutant forms (Dataset EV3), supporting the conclusion that R443 does not act through classical redox mechanisms. Together, these results suggest that R443 fulfills a structural role during morphogenesis, independent of its conserved thioredoxin motif, and is another example of exaptation of metabolic enzymes.

## Insights into uncoating of mimivirus nucleoid revealed by the conserved L410 protein

While the uncoating mechanism is well characterized structurally, the precise timing of core release into the cytoplasm remains unresolved. Taking advantage of the endogenously tagged version of L410, a highly abundant and conserved protein localized to the viral nucleoid, we decided to investigate this key step during mimivirus infection cycle. *Acanthamoeba castellanii* cells were infected at high multiplicity of infection (MOI) and processed for immunofluorescence microscopy at defined infection times: 0, 0.25, 0.5, 1, 3, and 5 hpi (Fig. S7). At early time points (0 and 15 min pi), no clear signal for L410 was detected, suggesting that the viral nucleoid remained enclosed within intact capsids, inaccessible to antibody detection. A small subset of cells (1%) displayed faint L410 puncta at 15 min, indicating that uncoating might begin in a limited number of particles. At 30 min pi, a substantial increase in L410 signal was observed, with ~33% of infected cells displaying discrete puncta, typically representing one viral core per cell. This proportion increased markedly at 1 hpi, with ~65% of cells showing multiple L410 puncta (1 to 10 per cell), consistent with progressive uncoating and release of additional cores into the cytoplasm (Fig. 6).

Interestingly, the frequency of detectable L410 signal decreased again at 3 hpi (~12% of cells), but reappeared at 5 hpi, now colocalized with early-stage VFs (Figs. 6 and S7). This biphasic pattern suggests an early wave of nucleoid detection corresponding to its release from the capsid into the cell cytoplasm, followed by a later stage of L410 de novo synthesis and recruitment to sites of viral assembly. We interpret the reduced detection at 3 h as the consequence of nucleoid disassembly and possible degradation, dispersal of L410, or epitope masking inside the VF.

Collectively, these results defined a temporal window for nucleoid release into the cytoplasm between 15 and 60 min pi, establishing this phase as a key inflection point in mimivirus replication cycle. Our observations on the spatiotemporal dynamics of the L410 protein during mimivirus infection are consistent with the nucleoid uncoating kinetics reported for the prototypical

poxvirus, vaccinia virus, where the nucleoid delivery strictly precedes viral genome release, typically around 60 min pi (Cyrklaff et al, 2007).

## Conclusion

We report a comprehensive interaction network of the mimivirus virion that enables the disentanglement of morphogenesis modules, offering an unprecedented view into the structural organization and complexity of its particles. Moreover, we establish a robust platform for probing protein conservation based on structure prediction in giant viruses, thereby enhancing the detection of potentially conserved functions. Our webserver, GV-Clusters (https://www.igs.cnrs-mrs.fr/gvclusters), will allow the community to browse our Virome+GV orthogroups, corresponding to an extension of the virome database (Nomburg et al, 2024) by more than 46,500 AF-predicted structures in the *Nucleocytoviricota* phylum. To facilitate broader use, we share the code of our clustering pipeline (https://src.koda.cnrs.fr/igs/gvcluster), allowing the community to access a structure-informed deep evolutionary history of their own datasets.

Our data support a model in which PPI networks are highly connected within compartments but only present few connections between the membrane network and the internal compartment. Similar results have recently been reported in *Marseilleviridae* particles (Mühlberg et al, 2025). Reconciling this independence with the tight and efficient coordination of virion assembly during morphogenesis emerges as a central question for understanding the logic of giant virus evolution.

## Methods

### Reagents and tools table

| Reagent/resource | Reference or source | Identifier or catalog number |
| --- | --- | --- |
| **Experimental models** | | |
| *Acanthamoeba castellanii* Neff | Douglas | 30010 |
| Acanthamoeba polyphaga mimivirus | Raoult et al, 2004 | |
| Mimivirus reunion | Villalta et al, 2022 | |
| **Recombinant DNA** | | |
| vAS1:3xHA-3'UTRmg105 -pmg153-Nourseothricin (HA-tag) | Philippe et al, 2024 | 198745 |
| vHB47: pmg741-Nourseothricin (KO) | Philippe et al, 2024 | 198747 |
| vAS1-NEO3xHA-3' UTRmg105 -pmg153-Geneticin (cis complementation) | This study | |
| vAS1-L515 | This study | |
| vAS1-L330 | This study | |
| vAS1-R710 | This study | |
| vAS1-R443 | This study | |
| vAS1-R387 | This study | |
| vAS1-L274 | This study | |
| vAS1-R347 | This study | |

| Reagent/resource | Reference or source | Identifier or catalog number |
| --- | --- | --- |
| vAS1-R287 | This study | |
| vAS1-R513b | This study | |
| vAS1-L567 | This study | |
| vAS1-L487 | This study | |
| vAS1-L323 | This study | |
| vAS1-R335 | This study | |
| vAS1-R317 | This study | |
| vAS1-L446 | This study | |
| vAS1-L593 | This study | |
| vAS1-R595 | This study | |
| vAS1-L410 | This study | |
| vAS1-L454 | This study | |
| vAS1-L264 | This study | |
| vAS1-R721 | This study | |
| vHB47-R443 | This study | |
| vAS1-NEO3xHA-3' UTRmg105-R443 complementation | This study | |
| vAS1-NEO3xHA-3'U TRmg105-R443 mutation nucleophilic cys | This study | |
| vAS1-NEO3xHA-3'UTRmg 105-R443 mutation resolving cys | This study | |
| **Antibodies** | | |
| HATag Recombinant Rabbit Monoclonal Antibody (RM305) | Invitrogen | MA5-27915 |
| HA Tag Monoclonal antibody (2-2,2,14) | Invitrogen | 11553060 |
| Goat Anti-Mouse IgG H&L (Alkaline Phosphatase) | Abcam | AB97020 |
| Donkey anti-Rabbit IgG (H + L) Highly Cross-Adsorbed Secondary Antibody, Alexa Fluor™ 594 | Invitrogen | A-21207 |
| **Oligonucleotides and other sequence-based reagents** | | |
| PCR primers | This study | Table EV1 |
| **Chemicals, enzymes, and other reagents** | | |
| Pierce Anti-HA Magnetic beads | Thermo Scientific | 88836 |
| Nitrocellulose membrane 0,45 µm | Bio-Rad | 1620251 |
| Power Blotter 1-Step (Transfer buffer 5x) | Invitrogen | PB7300 |
| Complete Mini EDTA-Free | Roche | 11836170001 |
| BCIP/NBT Liquid Substrate System | Sigma | B1911-100ML |
| Phusion™ High-Fidelity DNA Polymerase | Thermo Fisher | F530L |
| In Fusion Snap Assembly | Takara | 638948 |
| Terra PCR Direct RedDye Premix | Takara | 639286 |
| GoTaq | Promega | M3008 |

| Reagent/resource | Reference or source | Identifier or catalog number |
|---|---|---|
| NotI-HF | New England Biolabs (NEB) | R3189S |
| EcoRI-HF | New England Biolabs (NEB) | R0101S |
| HindIII-HF | New England Biolabs (NEB) | 3104S |
| SsoAdvanced Universal SYBR® Green Supermix | Bio-Rad | 1725274 |
| Prestained Protein ladder | Euromedex | 06P-0111 |
| Cesium Chloride | Euromedex | EU0770 |
| Proteose–peptone | Sigma-Aldrich (Merck) | 82450 |
| Gucose | Sigma | G8270-1KG |
| Yeast extract | Gibco | 288620 |
| Tris HCl | Euromedex | EU0011 |
| 16% Formaldehyde (w/v), Methanol-free | Fischer Scientific | 28906 |
| Polyfect | QIAGEN | 301107 |
| Nourseothricin | Jena Bioscience | AB-102L |
| Geneticin G418 | Thermo Fischer Scientific | 10131027 |
| Vectashield plus with DAPI 10 ml | Eurobio Scientific | H-2000-10 |
| **Software** | | |
| ApE plasmid-editing software | M. Wayne Davis | https://jorgensen.biology.utah.edu/wayned/ape/ |
| ImageJ | NIH | https://imagej.net/software/fiji/downloads |
| DotDotGoose | American Museum of Natural History, Center for Biodiversity and Conservation | https://biodiversityinformatics.amnh.org/open_source/dotdotgoose/ |
| LocalColabFold 1.5.2 | Mirdita et al, 2022 | https://github.com/YoshitakaMo/Localcolabfold |
| Foldmason | Gilchrist et al, 2026 | https://search.foldseek.com/foldmason |
| Cytoscape 3.10.1 | Shannon et al, 2003 | https://cytoscape.org/ |
| GOATOOLS | Klopfenstein et al, 2018 | https://github.com/tanghaibao/goatools |
| Mmseqs2 b22d5f6d02cb27ebc2cd931d8d20fe92ff54b8a8 | Steinegger and Söding, 2017 | https://github.com/soedinglab/MMseqs2 |
| Foldseek 07932751e776dd71b224dadc94aea0922d08e653 | Van Kempen et al, 2024 | https://github.com/steineggerlab/foldseek |
| FATCAT 2.0 | Li et al, 2020 | https://fatcat.godziklab.org/ |
| RCSB Pairwise alignment | Bittrich et al, 2024 | https://www.rcsb.org/alignment |
| **Other** | | |
| Monarch Plasmid Miniprep Kit | New England Biolabs (NEB) | T1010S |
| Monarch PCR and DNA clean up | New England Biolabs (NEB) | T1030S |
| Pure link genomic DNA mini kit | Thermo Fisher | K182001 |

| Reagent/resource | Reference or source | Identifier or catalog number |
|---|---|---|
| Flasks T175 | SARSTEDT | 833912002 |
| Hard-Shell® 96-Well PCR Plates, low profile, thin wall, skirted, clear | Bio-Rad | HSP9601 |

## Viral and cell strains

The viral strain used was Acanthamoeba polyphaga mimivirus (Raoult et al, 2004), and *Acanthamoeba castellanii* (Douglas) Neff (American Type Culture Collection 30010™) as the host.

## Endogenous HA-tagging of mimivirus genes by homologous recombination

The vAS1 plasmid (3xHA-pmg153-Nourseothricin) was used for endogenous tagging of mimivirus genes via homologous recombination (Philippe et al, 2024). For each target gene (L515, L330, R710, R443, R387, L274, R287, R347, L446, R335, L567, R513b, R317, L593, L323, R595, L410, R721, L264, and L454), two 500 bp homology arms were designed and amplified using gene-specific primers (HS1–HS67 and HB831–HB1047) shown in Table EV3, and introduced at the 5′ and 3′ end of the selection cassette to promote recombination with the viral DNA. Cloning steps were performed using Phusion Taq polymerase (ThermoFisher) and the In-Fusion system (Takara). Before transfection into *Acanthamoeba castellanii*, plasmids were linearized with restriction enzymes EcoRI and NotI (New England Biolabs).

## Knockout and cis-complementation of the thioredoxin domain-containing protein R443

The vHB47 plasmid (pmg741-NAT), containing a nourseothricin (NAT) resistance cassette, was used to perform the knockout of the R443 gene via homologous recombination (Philippe et al, 2024; Alempic et al, 2024). Two 500 bp homology arms flanking the coding R443 gene sequence were PCR-amplified from viral genomic DNA using Phusion High-Fidelity DNA Polymerase (ThermoFisher) and primers HS68–HS71 (Table EV3). The 5′ and 3′ arms were then cloned into vHB47 using the In-Fusion system (Takara) to flank the selection cassette and promote recombination with the viral genome. Before transfection into *Acanthamoeba castellanii*, the knockout construct was linearized by digestion with HindIII and NotI.

For cis-complementation, the NAT resistance cassette of the 3xHA-pmg153-NAT (vAS1) vector was replaced with a geneticin (neomycin) resistance cassette using In-Fusion cloning (Takara) and cassette-specific primers (Table EV3). The full-length wild-type (WT) R443 coding sequence, including its endogenous promoter, was inserted into the modified vector to generate the complementation construct (R443_WT) using the primers HB1200-HB1201. Site-directed mutagenesis of the catalytic CXXC motif (Appendix Fig. S6A) was performed to generate point mutants: R443_SXXC (mutation of the nucleophilic cysteine) and R443_CXXS (mutation of the resolving cysteine), using primers HB1202–HB1205 in which cysteine codons were substituted with serine codons. All

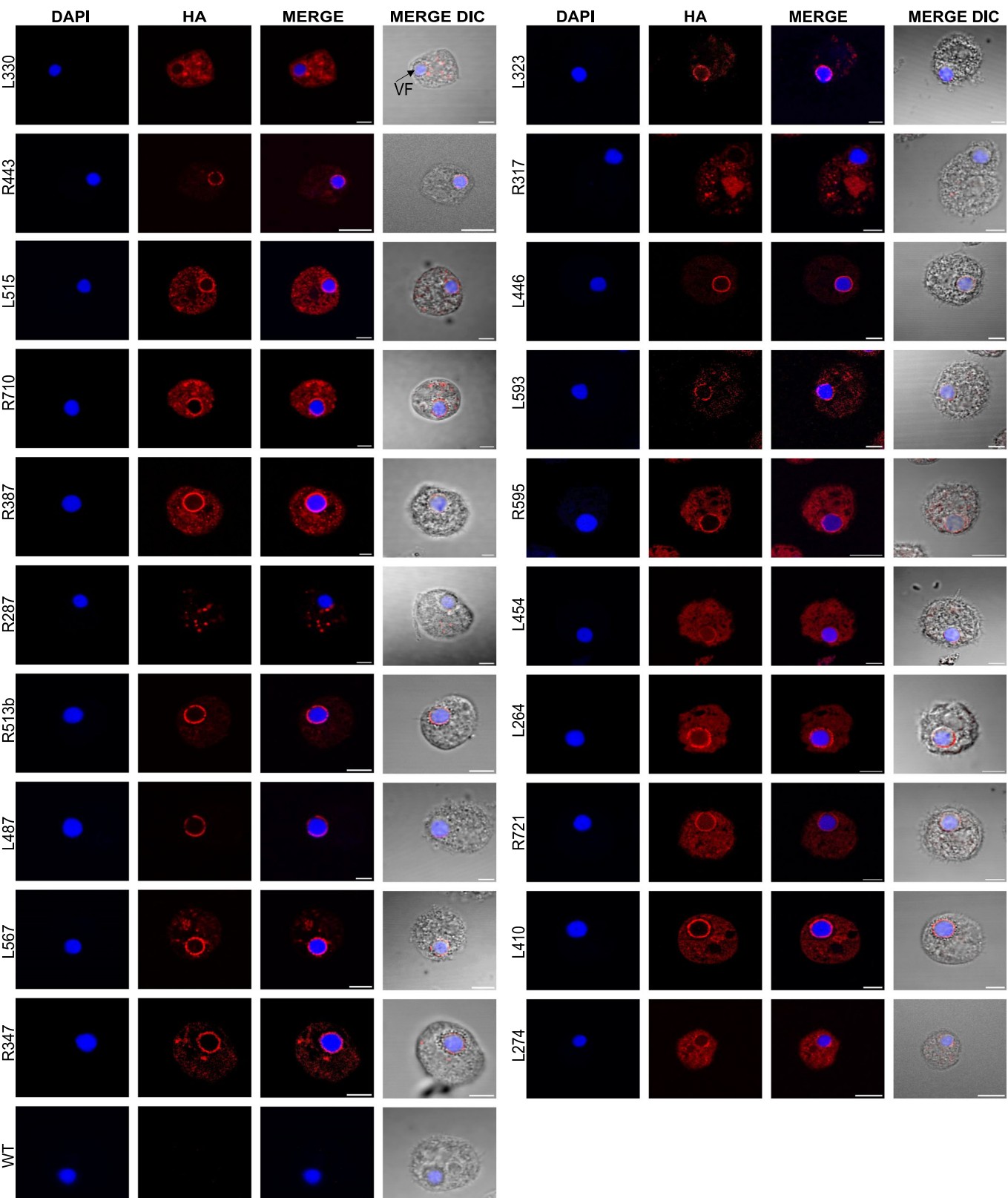

◀ **Figure 2. Mimivirus late proteins are mainly localized at the outer layer of the VF.**

Immunofluorescence (IF) images of mimivirus proteins endogenously tagged with C-terminal 3xHA. VFs were labeled with DAPI. Wild-type mimivirus-infected cells were included as IF control. Scale bar: 5 μm. IF was carried out on Acanthamoeba cells 6 hpi (MOI = 10). Source data are available online for this figure.

complementation vectors were linearized with HindIII and NotI before transfection into *A. castellanii*.

## Cell culture

*Acanthamoeba castellanii* cells were cultured in a custom PPYG medium consisting of 2% (w/v) proteose peptone, 0.1% yeast extract, 100 μM glucose, 4 mM $MgSO_4$, 0.4 mM $CaCl_2$, 50 μM $Fe(NH_4)_2(SO_4)_2$, 2.5 mM $Na_2HPO_4$, and 2.5 mM $KH_2PO_4$, adjusted to pH 6.5. The medium was supplemented with antibiotics (ampicillin 100 μg/mL and kanamycin 25 μg/mL). Cells were maintained for several days until reaching confluence.

## Cell transfection and mutant virus generation

*Acanthamoeba castellanii* cells were transfected with 6 μg of each linearized recombinant plasmid using SuperFect (QIAGEN) in phosphate saline buffer (PBS), according to the manufacturer's instructions. After 1 h, PBS was replaced with 2 ml PPYG medium, and cells were infected with wild-type mimivirus to generate HA-tagged or R443 knockout mutants. Nourseothricin (100 μg/ml) was used for selection. For cis-complementation of the R443 knockout, transfected cells were infected with the R443 KO mutant and selected with Geneticin (100 μg/ml). Cloning and genotyping were performed as described (Philippe et al, 2024). Recombinant viruses were produced and purified by cesium chloride (CsCl) gradient centrifugation as previously reported (Bisio et al, 2023).

## Competition assay and quantitative PCR analysis

For the competition assay, $7 \times 10^6$ *Acanthamoeba castellanii* cells were infected with a 1:1 mixture of wild-type mimivirus and R443 knockout virus (P0) at a multiplicity of infection (MOI) of 1. One hpi, cells were washed to remove extracellular virions. At 24 hpi, newly produced viruses (P1) were collected and used to infect fresh cells. This procedure was repeated for ten sequential passages (P1 to P10). For quantitative PCR, viral genomic DNA from each passage was extracted using the PureLink Genomic DNA Kit (Invitrogen). Primer pairs CG69–CG70 and CG71–CG72 (Table EV3) were used to amplify fragments of the NAT selection cassette and a mimivirus internal reference gene (R565 putative glutamine synthetase), respectively. PCR reactions were performed as described previously (Philippe et al, 2024). A standard curve was generated for each experiment using genomic DNA from purified R443 KO virus. Each data point represents the average of three biological replicates. Quantitative real-time PCR (qPCR) was performed on a CFX96 Real-Time System (Bio-Rad) using the SsoAdvanced™ SYBR® Green Supermix 2X (Bio-Rad). EvaGreen fluorescence was measured at the end of each cycle to monitor amplification kinetics. Threshold cycle (Ct) values from technical triplicates were averaged and used for relative DNA quantification, as detailed in (Philippe et al, 2024).

## Immunofluorescence localization of HA-tagged mimivirus proteins

*A. castellanii* cells were seeded on poly-L-lysine-coated coverslips in 12-well plates and infected with each HA-tagged mimivirus mutant at a MOI of 20. Cells infected with wild-type mimivirus served as a negative control. One hpi, cells were washed with PBS to remove unbound virions. At 6 hpi, cells were fixed with 4% paraformaldehyde (PFA) in PBS for 20 min at room temperature. Immunofluorescence staining was performed as previously described (Philippe et al, 2024). Briefly, fixed cells were incubated with rabbit anti-HA primary antibody (Invitrogen, 1:1000) followed by Alexa Fluor 594-conjugated goat anti-rabbit secondary antibody (Invitrogen, 1:1500). Coverslips were mounted onto glass slides using 3.5 μl of VECTASHIELD mounting medium with DAPI (Eurobio Scientific). Imaging was performed on an Olympus FV1000 confocal microscope using a ×100 objective lens and a ×1.6 Optovar for DIC, DAPI, or Alexa Fluor 594 fluorescence acquisition.

## Co-immunoprecipitation (co-IP) and western blot (WB) analysis

Immunoprecipitation of HA-tagged mimivirus proteins was performed as previously described (Klockenbusch and Kast, 2010), with minor modifications. Briefly, confluent *A. castellanii* cells were infected with genetically modified mimivirus expressing HA-tagged bait proteins at MOI = 10. Control cells were infected with wild-type mimivirus. One hpi, the medium was replaced with fresh PPYG to remove extracellular virions. At 6 hpi, cells were harvested by centrifugation at 900×*g* for 5 min. For cross-linking, cell pellets were resuspended in 4% formaldehyde and incubated at room temperature (RT) for 10 min with gentle agitation. Cells were then pelleted (900×*g*, 5 min, RT) and cross-linking was quenched by adding ice-cold 1.25 M glycine. After an additional centrifugation, pellets were washed twice in PBS and stored at −80 °C. For lysis, frozen pellets were resuspended in RIPA buffer (50 mM Tris-HCl, 150 mM NaCl, 1% IGEPAL, 0.5% sodium deoxycholate, 0.1% SDS, 1 mM EDTA, protease inhibitors [Complete Mini, EDTA-free, Roche], adjusted to pH 8.0). Lysates were subjected to five freeze–thaw cycles followed by sonication for 10 min. Debris was removed by centrifugation at 20,000×*g* for 15 min at 4 °C. Supernatants containing HA-tagged proteins were incubated for 2 h with pre-equilibrated anti-HA magnetic beads (Thermo Fisher). Beads were washed three times with RIPA buffer and resuspended in Laemmli buffer (without reducing agent) for WB or mass spectrometry analysis. Detection of HA-tagged proteins by WB was performed as described in (Philippe et al, 2024), using a mouse monoclonal anti-HA antibody (Invitrogen, 1:1000) and goat anti-mouse IgG H&L secondary antibody (Abcam, 1:2000). Biological triplicates for each bait and control condition were prepared and submitted for mass spectrometry.

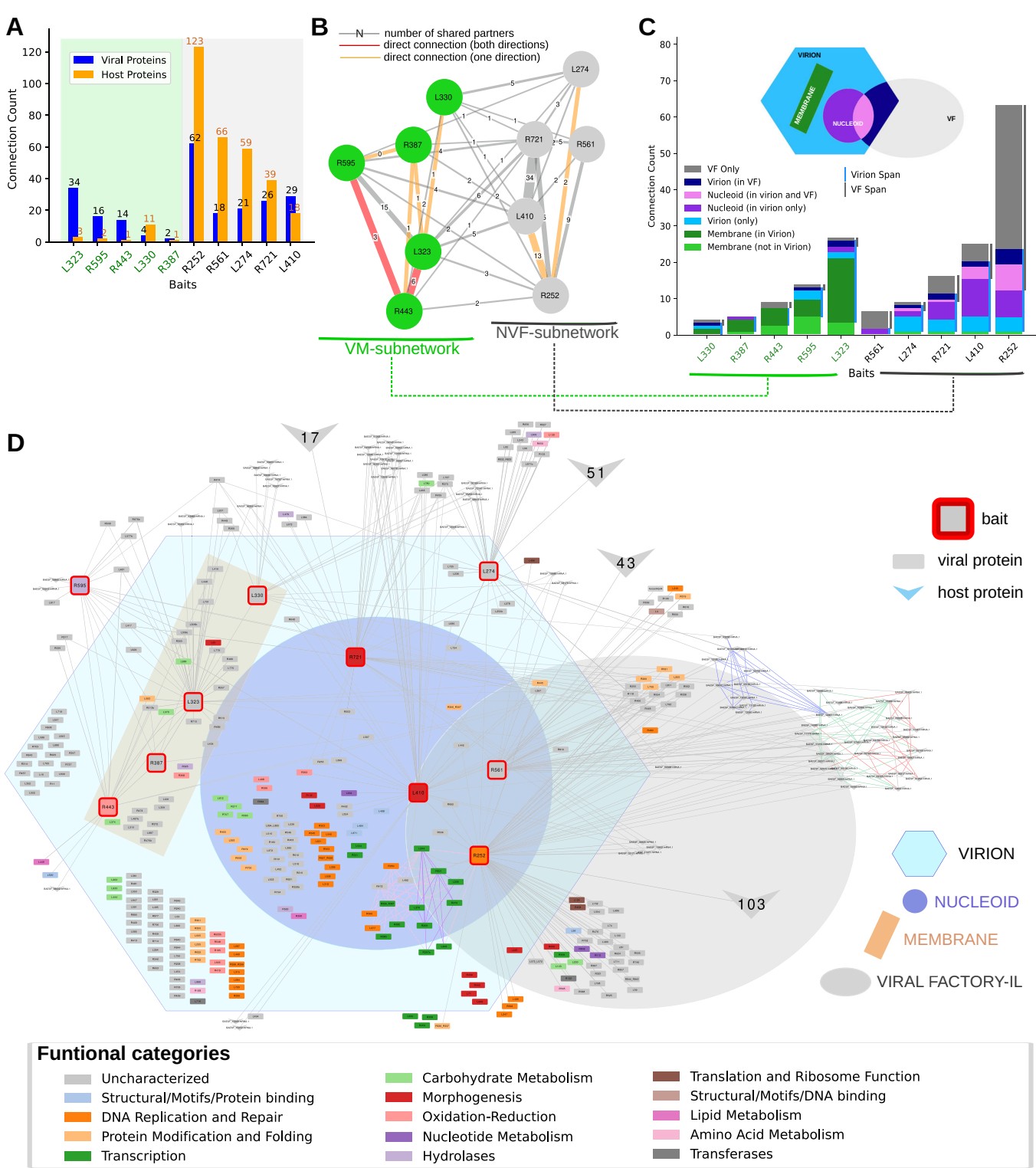

**Funtional categories**

| | | |
|---|---|---|
| Uncharacterized | Carbohydrate Metabolism | Translation and Ribosome Function |
| Structural/Protein binding | Morphogenesis | Structural/Motifs/DNA binding |
| DNA Replication and Repair | Oxidation-Reduction | Lipid Metabolism |
| Protein Modification and Folding | Nucleotide Metabolism | Amino Acid Metabolism |
| Transcription | Hydrolases | Transferases |

## Quantification of major core protein (L410) puncta in infected cells

*A. castellanii* cells were seeded on poly-L-lysine-coated coverslips in 12-well plates and infected with the HA-tagged L410 mutant at MOI = 10. Cells were synchronously infected and fixed at

defined time points pi (0, 0.25, 0.5, 1, 3, and 5 h). Following fixation, immunolabeling, and DAPI staining, images of at least 120 cells per time point were acquired using a Zeiss Axio Observer Z1 inverted microscope. Quantification of L410 puncta was performed using the freely available image analysis software DotDotGoose. Puncta counts were normalized per cell and

◄ **Figure 3. The co-IP network.**

(A) Count of viral (blue) and host (orange) proteins identified as connected to each bait by IP-MS. The first 5 baits (green background) corresponding to membrane proteins are mostly connected to viral proteins. (B) Summary of the IP-MS baits network, based on the number of common connected proteins (edge label). Direct connections between bait proteins are indicated in red (observed in both directions) or in orange (only one direction). Two sub-networks are identified by Louvain clustering, the first contains the membrane proteins (VM subnetwork, green nodes), and the second cluster is shown with gray nodes (NVF subnetwork). (C) For each bait, the number of connected proteins is shown by region (green: Membrane, blue: Virion, violet: Nucleoid). Baits with transmembrane regions (green label: L330, R387, R443, R595, and L323) are connected preferentially within the VM subnetwork. Nucleoid-VF subnetwork (R561, L274, R721, L410, R252) present preferential connections to the NVF subnetwork, which are indicated by colored edges. (D) Detailed network showing all connections (gray edges) of the baits (red border nodes) to their connected proteins. Background-colored regions illustrate the protein nodes localization and/or the membrane proteins (Virion, Nucleoid, VF-IL, Membrane). Functional categories are depicted as node colors. Direct protein-interaction subnetworks, obtained by homology with PDB complexes are indicated by colored edges (Violet/Pink: 6rfl, 7amv (poxvirus), 8q3k (asfv), 7und, 7nvu, 7ena (human) RNA polymerase complex, Blue: 7dvq Human Minor Spliceosome complex, Green: 7a09 human ABCE1-bound 43S pre-initiation complex, Red: 6zj3 atypical cytoplasmic ribosome of *Euglena gracilis*). Source data are available online for this figure.

analyzed across time points to assess L410 localization during infection.

## MS-based quantitative proteomic analyses

Four independent IP-MS experiments were analyzed: R387-HA and WT1 (Exp.1), L330-HA, R443-HA and WT2 (Exp.2), L323-HA, R595-HA, L410-HA, R721-HA, L274-HA and WT3 (Exp.3), and R443_WT-HA, R443_SXXC-HA and R443_CXXS-HA (Exp.4). Co-purified proteins, resuspended in Laemmli buffer, were stacked in the top of a NuPAGE 4–12% gel (Invitrogen) and stained with Coomassie Blue R-250 (Bio-Rad). They were then in-gel digested using modified trypsin (Promega, sequencing grade) as previously described (Casabona et al, 2013), except that Tris(2-carboxyethyl) phosphine hydrochloride was used instead of dithiothreitol. The resulting peptides were analyzed by online nanoliquid chromatography coupled to MS/MS (Ultimate 3000 RSLCnano and Q-Exactive HF, Thermo Fisher Scientific, for Exp.1, Exp.2 and Exp.4, and UHPLC Vanquish Neo and Orbitrap Ascend Tribrid, Thermo Fisher Scientific, for Exp.3). Peptides were sampled on a precolumn (300 µm x 5 mm PepMap C18, Thermo Scientific) and separated in a 75 µm × 250 mm C18 column (Aurora Generation 3, 1.7 µm, IonOpticks) using an acetonitrile gradient of 50 min for Exp.1 and Exp.2 and 80 min for Exp.3 and Exp.4. The MS and MS/MS data were acquired by Xcalibur version (Thermo Fisher Scientific).

Peptides and proteins were identified by Mascot (version 2.8, Matrix Science) through concomitant searches against the home-made mimivirus and *Acanthamoeba castellanii* databases, and a homemade database containing the sequences of classical contaminant proteins found in proteomic analyses (keratins, trypsin, etc.). Trypsin/P was chosen as the enzyme, and two missed cleavages were allowed. Precursor and fragment mass error tolerances were set at 10 and 20 ppm, respectively. Peptide modifications allowed during the search were: Carbamidomethyl (C, fixed), Acetyl (Protein N-term, variable) and Oxidation (M, variable). The Proline software ((Bouyssié et al, 2020), version 2.3) was used for the compilation, grouping, and filtering of the results: conservation of rank 1 peptides, peptide length ≥6 amino acids, false discovery rate of peptide-spectrum-match identifications <1% (Couté et al, 2020), and minimum of one specific peptide per identified protein group. Proline was then used to perform a MS1-based label-free quantification of the identified protein groups.

Statistical analyses were performed using Prostar (Wieczorek et al, 2017). Proteins identified in the contaminant databases or quantified in less than three replicates of one condition were discarded. After log2 transformation, abundance values were normalized using the variance stabilizing normalization (vsn) method for Exp.1, Exp.2 and Exp.3, or the median abundance value of the bait protein for Exp.4. Missing values were then imputed, using the SLSA algorithm for partially observed values in the condition and the DetQuantile algorithm for totally absent values in the condition. Statistical testing was conducted with limma (Ritchie et al, 2015), whereby differentially expressed proteins were selected using a log2(fold change) ≥1 and a *P* value < 0.01, allowing to reach a false discovery rate inferior to 5% according to the adaptive Benjamini–Hochberg estimator. Proteins found differentially abundant but quantified in less than three replicates in the condition in which they were found to be more abundant were manually invalidated (*P* value = 1).

## Statistics and reproducibility

Data are represented as the mean of three independent biological replicates per experiment (*n* = 3, unless stated otherwise). The error bars are representative of the standard deviation from the mean for each experiment.

## Mimivirus structural information available from different sources

The count of mimivirus proteins with direct or homologous information available in the PDB database was retrieved by searching the PDB sequences with the mimivirus protein sequences as queries using Blastp (*e* value < 1e-9, coverage >90%). The count of mimivirus proteins showing AF models with at least an intermediate confidence level (pLDDT>70) was assessed.

## Structural AlphaFold based clustering of the enhanced virome dataset

We have enhanced the (Nomburg et al, 2024) virome dataset (4642 genomes) with 62 additional genomes from the GVDB (Table EV1) relevant for this study, from the *Megaviricetes* (*Imitervirales*, *Pimascovirales*, *Pandoravirales*, *Algavirales* and *Mamonovirales*) and *Pokkesviricetes* (*Asfuvirales* and *Chitovirales*). These 62 genomes were folded using ColabFold 1.5.2 (Mirdita et al, 2022) thanks to the IDRIS Jean Zay GPU supercomputing facility. MSA computation was performed on the PACA bioinfo computing platform using MMseqs2 and Uniref30_2302 and EnvDB_202108 databases retrieved from the ColabFold webserver. The official ICTV taxonomy information was used for the additional genomes,

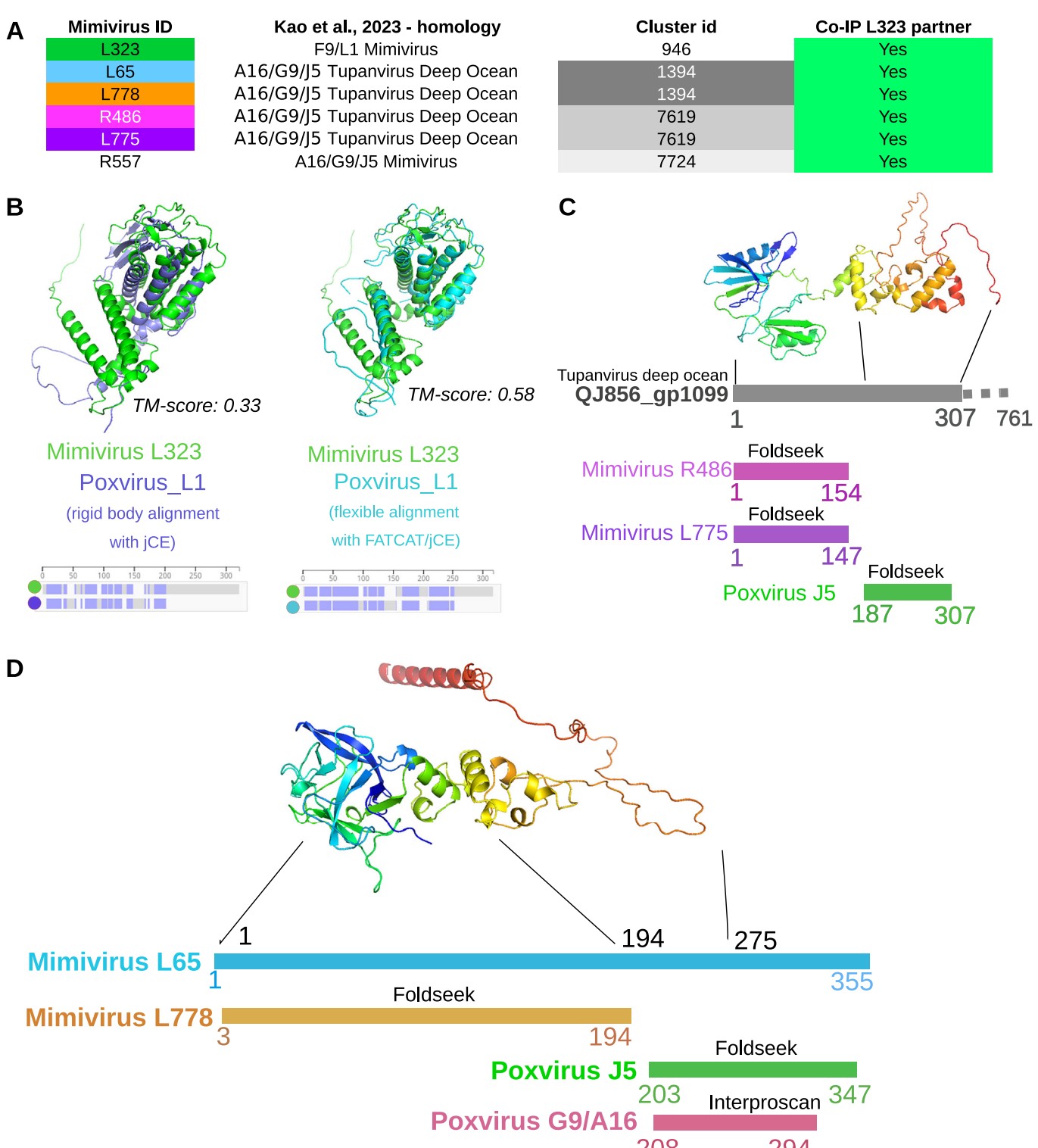

except for the Pandoraviruses taxonomy defined as in (Aylward et al, 2021), Medusavirus as in (Zhang et al, 2023), and Pacmanviruses as in (Matsuyama et al, 2023). The subfamily taxonomy, not used in the original Nomburg study, was added only for all genomes in the *Mimiviridae* family. The final dataset genome count is 4703, corresponding to 114,247 genes.

## Structural clustering merge step

Highly connected protein clusters were merged by structural similarity using an all-versus-all structural similarity search (Foldseek Tmscore>0.4, coverage >0.7), based on which clusters were subsequently merged if over 65% of proteins in one cluster had links to at

**Figure 4.  Entry fusion proteins in the membrane subnetwork (VM).**

(A) List of the poxvirus F9/L1 and A16/G9/J5 homologs in mimivirus (Kao et al, 2023). They are all partners of L323. (B) Structural alignment of AF models of mimivirus L323 and poxvirus L1 (VACWR088). Rigid body alignment (left) led to poor TM-score, due to poxvirus L1 model low pLDDT score. (right) Flexible alignment with FATCAT improved alignment score (TM-score=0.58). (C) Homolog of poxvirus J5 in tupanvirus deep ocean (QJ856_gp1099, (Kao et al, 2023)) shares its N-terminal domain with mimivirus (R486 and L775) and is absent in poxvirus EFC proteins, and might thus be specific to a putative undescribed EFC specific to *Mimiviridae*. R486 and L775 were both identified in the L323 IP-MS experiment. (D) Mimivirus L65 contains an Entry Fusion domain (homology with poxvirus J5) that was identified by Foldseek (between residues 203 and 347) and Interproscan (residues 208–294) extended by a N-terminal domain absent from poxvirus, but also present in mimivirus L778. Both L65 and L778 were identified in the L323 IP-MS experiment. Source data are available online for this figure.

least one member of another cluster. This threshold (65%) was set to be the most conservative possible, and still merged all 4 mimivirus clusters for which the minimum connected proteins percentage was 69% (96% for R357b, 96% for R382, 69% for R383 and 96% for L492). The mimivirus RNA polymerase subunits (Fig. EV1A,C; Table EV2) were determined from (Legendre et al, 2011).

## Estimating the LCA taxonomic level

As in Nomburg et al (Nomburg et al, 2024), the taxonomic level of the Last Common Ancestor was systematically estimated for each mimivirus protein-containing cluster. Taxonomy level was identified as the $N^{th}$ taxonomic level (e.g., *Megamimivirinae* subfamily) if all members have the same taxonomy at this level, and different taxonomies at lower levels (N-1, e.g., *Mimivirus* genus).

## Virion, nucleoid, and VF protein sets

The virion proteome was retrieved from Villalta et al from Mimivirus reunion (Villalta et al, 2022), and transferred to mimivirus by homology using MMseqs2-RBH (Reciprocal Best Hit), leading to 206 virion proteins. The nucleoid protein content was retrieved from purified nucleoids extracted from mimivirus reunion virions and submitted to MS-based proteomic analysis (Dataset EV4, Tabs 1-2). The mimivirus reunion nucleoids were obtained by treating purified virions with Proteinase K (1 mg/mL) for 1 h at 55 °C then overnight at 4 °C. The nucleoids were purified by ultracentrifugation and density gradients, first through a sucrose cushion (20% and 50% sucrose in Tris HCl 40 mM pH 7.5; 2 h at $133{,}907 \times g$ at 4 °C) then through a potassium tartrate gradient (10% to 80% potassium tartrate in Tris HCl 40 mM pH 7.5; 20 h at $268{,}000 \times g$ at 4 °C). The resulting band was harvested, and the nucleoids washed three times with Tris HCl 40 mM pH 7.5 (pelleted by centrifugation at $16{,}162 \times g$ for 5 min) and resuspended. The pellet and the supernatant fractions were prepared for MS-based proteomics, as described above. Pellet and supernatant of virions prepared in the same way were processed in parallel. Results obtained from nucleoid and virion pellets on one side and results obtained from nucleoid and virion supernatants on the other side were compared on the basis of spectral countings. Proteins were considered as potentially enriched in nucleoid fraction if they were identified with at least 2 peptides and identified only in nucleoid fraction or enriched at least 2 times in nucleoid fraction compared to virion fraction, either in pellet or supernatant. Results are presented in Dataset EV4. The resulting protein content was again transferred by homology to mimivirus using the same approach as for the virion proteome. As a result, 86 nucleoid proteins were considered in this study. The protein content of the internal layer of

the VF (VF-IL) was assessed using data from Rigou et al (Rigou et al, 2025), leading to 64 VF-Il viral proteins and 123 host proteins.

## Prediction of transmembrane domain-containing proteins

Phobius (Käll et al, 2004), DeepTMHMM (Hallgren et al, 2022), and TMBed (Bernhofer and Rost, 2022a) were used to predict transmembrane domains (TMD) in the mimivirus proteome. To reduce false-positive predictions, we considered a protein to be positive if TMD were predicted by at least two methods. Phobius identified 141 TMD proteins, TMBed identified 100 TMD, and DeepTMHMM identified 90 TMD proteins. While 82 proteins were positive for all three methods, the chosen combination method led to 100 proteins.

## Co-IP data analysis by subnetworks

A summary network of the co-IP baits has been built using Python scripting and displayed using Cytoscape (Shannon et al, 2003) (Fig. 3B), with edge weights defined based on the number of common connected proteins. Direct connection between bait proteins (when a bait is obtained in the IP-MS result of another bait) has been set to a higher edge weight as follows: edge_weight = Nc + 10*Ndc, with Nc the number of shared connected proteins between the two baits, and Ndc the number of direct connection (Ndc=1 or Ndc=2). Applying Louvain clustering has been achieved using the Python–Louvain community package (Blondel et al, 2008) and led to the identification of two sub-networks.

## Building a global PPI network with annotations

We used Cytoscape (Shannon et al, 2003) in order to build a global PPI network describing the co-IP experimental data. Nodes' positions were defined by their regions, inferred by previous proteomic data (see "Virion, nucleoid and VF protein sets", as well as "Prediction of transmembrane domain containing proteins sections"). Viral proteins are shown as rectangular nodes, and host proteins as V-shaped nodes. The host nodes were sometimes grouped in larger V-shaped nodes with protein counts.

## Host and viral proteins GO terms enrichment analysis

*A. castellanii* GO terms were retrieved from (Matthey-Doret et al, 2022). Mimivirus GO terms were retrieved through Uniprot50 homology (Blastp *e* value < 1e-5 and coverage>70%, Foldseek using mimivirus AF2 models versus AFDB-Uniprot50 and *e* value < 1e-5, coverage >70%). The Goatools python package (Klopfenstein et al,

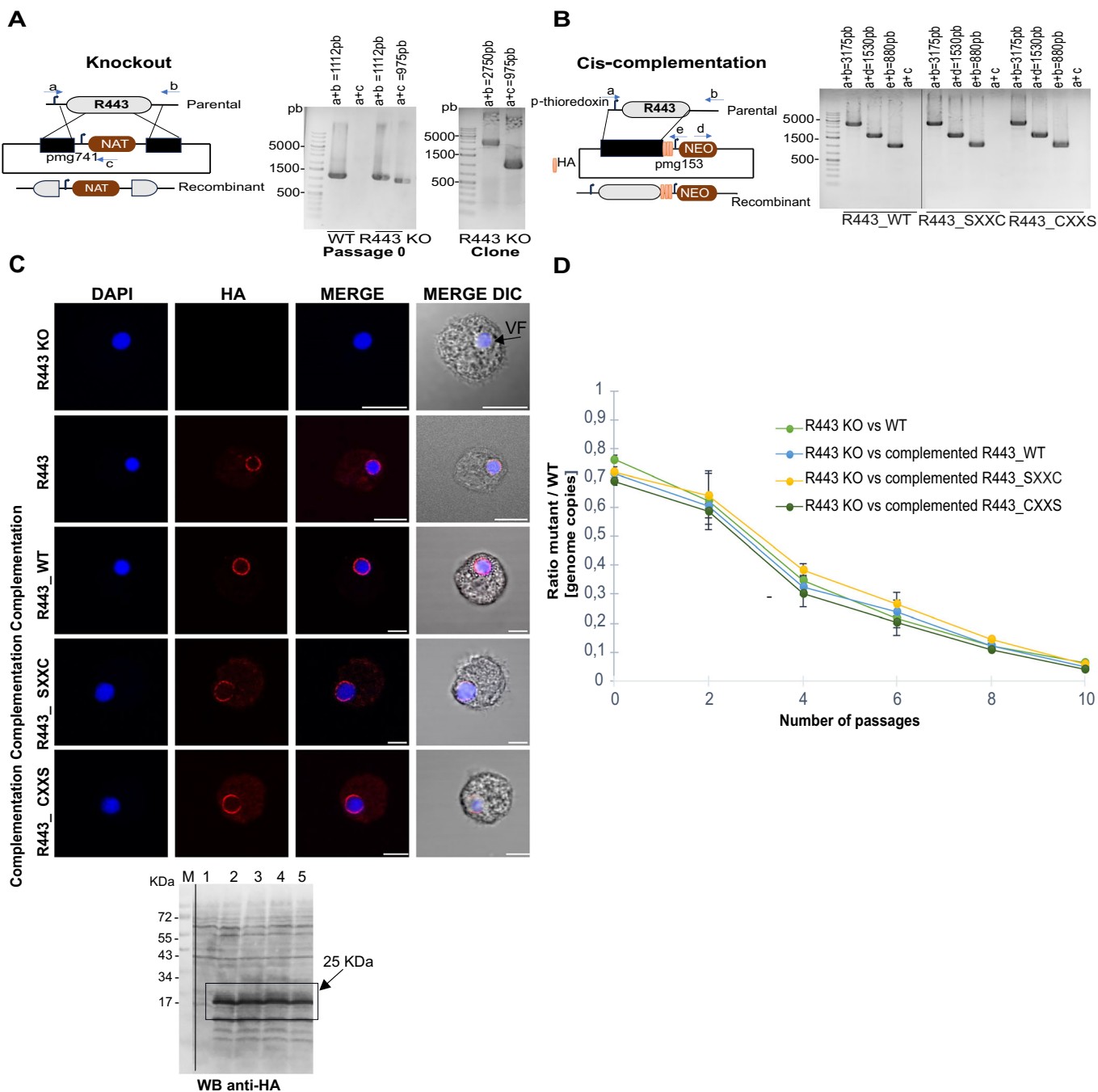

**Figure 5. Deletion of mimivirus thioredoxin domain-containing protein R443 impacts on viral replication and phenotype.**

(A) Schematic representation of the vector and strategy used for R443 Knockout. Recombinant viruses were generated by homologous recombination, selected using NAT selection cassette and cloned. NAT nourseothricin N-acetyl transferase. Clonality of the KO mutant was confirmed by PCR using the primers annealing a, b and c shown in the figure. Expected sizes are indicated in the figure (a + b for parental viruses and a + c for recombinant viruses genotyping). (B) Schematic representation of the vector and strategy used for R443 KO cis-complementation. Three recombinant complemented mutants R443_WT, R443 _SXXC and R443_CXXS were generated by homologous recombination and selected using neomycin (geneticin) selection cassette. Primers annealing locations a,b,d and e are shown and clonality of complemented mutants is confirmed by PCR. Expected sizes are indicated in the figure (a + b for parental viruses and a + d; e + b for recombinant viruses genotyping; a + c as negative control). (C) Immunofluorescence analysis at 6 hpi of R443 KO, R443 (HA-tagged virus as in Fig. 2, lane 2) and the complemented mutants R443_WT(CXXC), R443_SXXC and R443_CXXS endogenously tagged with 3×HA at the C-terminal. VFs were labeled with DAPI. Scale bar: 5 µm. Western blot analysis (anti-HA) corresponds to lanes 1–5, respectively: 1.R443 KO, 2. R443,3. R443_WT(CXXC),4. R443_SXXC,5. R443_CXXS. (D) Growth competition assay of R443 mutants. The competition was performed at MOI = 1 until passage 10. To evaluate the KO/WT ratio (copy number), qPCR analyses were performed using an endogenous locus (present in WT and recombinant viruses and the Nourseothricin selection cassette (only present in recombinant viruses). Data correspond to the mean ± SD of three independent experiments. Source data are available online for this figure.

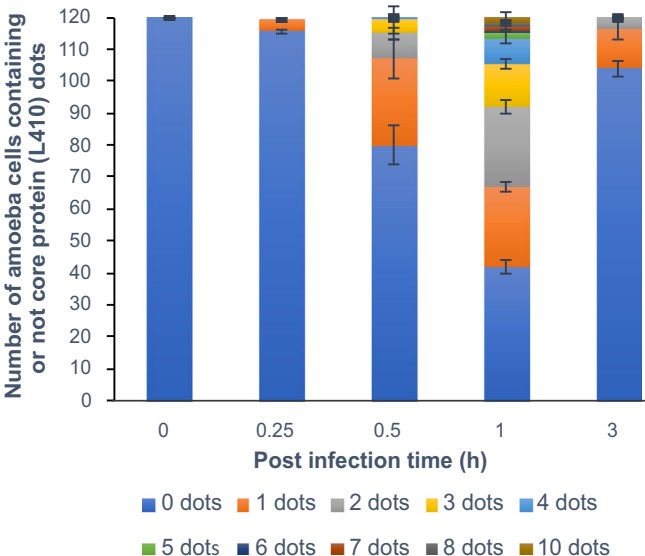

**Figure 6.  Quantification of amoeba cells displaying puncta of the L410 major core protein across infection stages.**

Quantification of amoeba cells containing 0 to 10 puncta of the L410 major core protein at different stages of infection (0, 0.25, 0.5, 1, 3, and 5 hpi). 120 amoeba cells were recorded per condition. Data correspond to the mean ± SD of three independent experiments. Source data are available online for this figure.

2018) was used to estimate GO enrichment for each co-IP bait partner set *versus* the whole Amoeba proteome. Enrichment was assessed using a Bonferroni-corrected *P* value < 0.01.

### Homologous PDB complexes search

Python pipeline was used, based on Mmseqs2 (*e* value < 0.01) and/or Foldseek (*e* value < 0.001) to identify complexes in the PDB database (v.5e639f2a516a2e77ad86f21ec9f30ad9 from 2023-04-20) sharing the highest number of homologous chains. Previously computed Alphafold2 (AF2) models were used for structural searches. To identify the largest complexes and avoid redundancy, all hits were sorted by their number of homologous chains and clustered using a similarity measure (cosine similarity) of the associated list of mimivirus homologous chains. Each of the three selected homologous PDB complexes was used to build direct interaction networks, connecting the identified chains, homologous to mimivirus proteins, if their interface surfaces distances less than 8 Å.

RNA polymerase PDB complexes of poxvirus and ASFV (6RFL, 7AMV, 8Q3K) were used to build putative direct interaction networks of mimivirus RNA polymerase complex. Known mimivirus homologs were connected (edge of this putative network) if their homologous subunits were closer than 8 Å.

### Expression clusters network

Temporal expression patterns were retrieved from Bessenay et al (Bessenay et al, 2024) for Megavirus chilensis and transferred to mimivirus by using Mmseqs2-RBH mapping. It consists of five expression clusters (GV-1 corresponds to the earliest expression, and GV-5 for the latest expression group).

### Icosahedral shape enrichment in different regions of the virion

We extended our analysis of the statistical association between viral shape and protein composition across different virion regions (nucleoid, membrane, and other virion proteins) (Appendix Fig. S4), using additional proteomic datasets from four icosahedral-shaped viruses belonging to three families: *Mimiviridae*, *Marseilleviridae*, and *Asfarviridae*. We used mimivirus reunion nucleoid content from this study, and virion proteome from Villalta et al (Villalta et al, 2022), tupanvirus nucleoid content from Schrad et al (Schrad et al, 2020), and virion proteome from Abrahao et al (Abrahão et al, 2018), melbournevirus nucleoid and virion content from Mühlberg et al (Mühlberg et al, 2025), and African swine fever virus nucleoid and virion content from (Alejo et al, 2018). For this specific analysis, membrane-associated proteins were predicted using TMBed (Bernhofer and Rost, 2022a) for transmembrane domain prediction. For each of the four icosahedral-shaped viruses, we estimated the odds ratio and corresponding *P* value using Fisher's exact test (Appendix Fig. S4; Dataset EV5, Tabs 1–4). The test was applied to contingency matrices comparing icosahedral or mixed morphologies with membrane or nucleoid regions, as well as membrane versus other virion regions.

### Annotation of Pacmanvirus lupus

The genome sequence of Pacmanvirus lupus (Alempic et al, 2023) was submitted to GeneMark v4.32 with the virus option to predict 506 genes. Corresponding protein sequences were submitted to blastp v2.17.0 (Altschul et al, 1990) against swissprot (May 2025, (Trgovec-Greif et al, 2024)) and the clustered version of the NCBI nr database (May 2025) with an *e* value of $1e^{-2}$ and $1e^{-5}$, respectively. The VOG database (September 2025,) was screened using hmmer v3.4 (Eddy, 2011). Interproscan v5.75-106.0 (Jones et al, 2014) was used to perform a domain search against PANTHER v19.0, Pfam v37.4, SMART v9.0, ProSiteProfiles v2025_01, ProSitePatterns v2025_01 and Hamap v2025_01 databases. The online version of CDSearch at NCBI was used with default parameters to screen the conserved domain database (Marchler-Bauer et al, 2015). Potential (trans)membrane proteins were predicted using TMBed v1.0 (Bernhofer and Rost, 2022b). These results were manually integrated to improve the functional annotation. The major capsid protein, spliced over 6 exons, was identified and assembled by homology to other viruses. Colabfold protein structure models of all remaining "hypothetical proteins" were submitted to foldseek v07932751e776dd71b224dad-c94aea0922d08e653 against the PDB database (*e* value < 0.1, proba >0.9, query coverage and subject coverage >60%) to find structural homologs. The complete genome sequence was scanned using tRNAscan-SE v2.0.12 (Chan and Lowe, 2019) with its default parameters to predict tRNA. No rRNA was predicted using RNAmmer v1.2 (Lagesen et al, 2007).

## Data availability

Predicted AlphaFold2 models for the GVDB can be downloaded as a compressed archive on Zenodo (https://doi.org/10.5281/zenodo.

17296301). For high-throughput searches, foldseek databases are also available on the same Zenodo repository for the GVDB and Virome+GVDB. Viewing and searching the Virome+GVDB clusters can be done through a dedicated webserver (https://www.igs.cnrs-mrs.fr/gvclusters). Co-IP networks with functional categories, LCA taxonomic level conservation, icosahedral shape status, and expression cluster color codes are also available on Zenodo (https://doi.org/10.5281/zenodo.17313812). The mass spectrometry proteomics data have been deposited to the ProteomeXchange Consortium (https://www.proteomexchange.org/) via the PRIDE (Perez-Riverol et al, 2022) partner repository with the dataset identifier PXD068780 (https://www.ebi.ac.uk/pride/archive/projects/PXD068780). The code of the pipeline used for the clustering computation in this study is available at https://src.koda.cnrs.fr/igs/gvcluster.

The source data of this paper are collected in the following database record: biostudies:S-SCDT-10_1038-S44318-026-00770-8.

## Peer review information

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

## Acknowledgements

We thank the members of the IMM imaging platform (Artemis Kosta and Hugo Le Guenno) and Yann Denis (IMM, Transcriptomics platform) for their technical assistance. Funded by the European Union (grant agreement No 832601 ERC-2018-ADG to CA, and grant agreement No. 101160452, ERC-2024-STG to H.B). Views and opinions expressed are, however, those of the authors only and do not necessarily reflect those of the European Union or the European Research Council (ERC). Neither the European Union nor the granting authority can be held responsible for them. The proteomic experiments were partially supported by Agence Nationale de la Recherche under projects ProFI (Proteomics French Infrastructure, ANR-10-INBS-08 & ANR-24-INBS-0015) and GRAL, a program from the Chemistry Biology Health (CBH) Graduate School of University Grenoble Alpes (ANR-17-EURE-0003). We thank the AMU MAB LIRMM platform for computing support and the GENCI-IDRIS for the GPU HPC resources used for AlphaFold predictions (grant 2022-AD011013526, 2023-AD011013526R1, and 2024-AD011013526R2).

## Author contributions

**Hela Safi**: Conceptualization; Data curation; Formal analysis; Investigation; Visualization; Methodology; Writing—original draft; Writing—review and editing. **Alain Schmitt**: Conceptualization; Data curation; Software; Formal analysis; Investigation; Visualization; Methodology; Writing—original draft; Writing—review and editing. **Alwena Tollec**: Formal analysis; Investigation. **Lucid Belmudes**: Formal analysis; Investigation. **Agathe M G Colmant**: Formal analysis; Validation; Methodology; Writing—review and editing. **Olivier Poirot**: Data curation; Software; Formal analysis; Visualization. **Sebastien Santini**: Data curation; Formal analysis. **Matthieu Legendre**: Data curation; Formal analysis; Investigation; Methodology; Writing—review and editing. **Yohann Couté**: Formal analysis; Validation; Methodology; Writing—review and editing. **Hugo Bisio**: Conceptualization; Supervision; Investigation; Writing—original draft; Writing—review and editing. **Chantal Abergel**: Conceptualization; Resources; Supervision; Funding acquisition; Writing—original draft; Writing—review and editing.

Source data underlying figure panels in this paper may have individual authorship assigned. Where available, figure panel/source data authorship is listed in the following database record: biostudies:S-SCDT-10_1038-S44318-026-00770-8.

## Disclosure and competing interests statement

The authors declare no competing interests.

# Expanded View Figures

**Figure EV1.   Mimivirus AF models in clusters containing poxvirus and ASFV RNA polymerase PDB complexes.**

(A) Poxvirus 6RFL RNA polymerase PDB complex and mimivirus homologous AlphaFold3 model. Only the 7 subunits with mimivirus homologs are shown. (B) Poxvirus 6RFL_J structure and superimposed mimivirus R357b AF model. Pymol cealign method led to an RMSD of 2.21 Å over 48 residues. (C) Summary table of the 6RFL and mimivirus homologous proteins with sequence or structure similarity e value. (D) Clustering merge step strategy illustrated by 6RFL_J and R357b clusters connections, based on structural similarity matches (Foldseek c >70% and TM-score>0.4) with > 65% of the other cluster. (E) Comparison of the LCA taxonomic levels for all clusters containing a mimivirus member, with or without the merge step. The merge step correction reduces the overall cluster count from 690 to 613, lowering the subfamily level cluster count while increasing the superkingdom taxonomic level counts.

## A

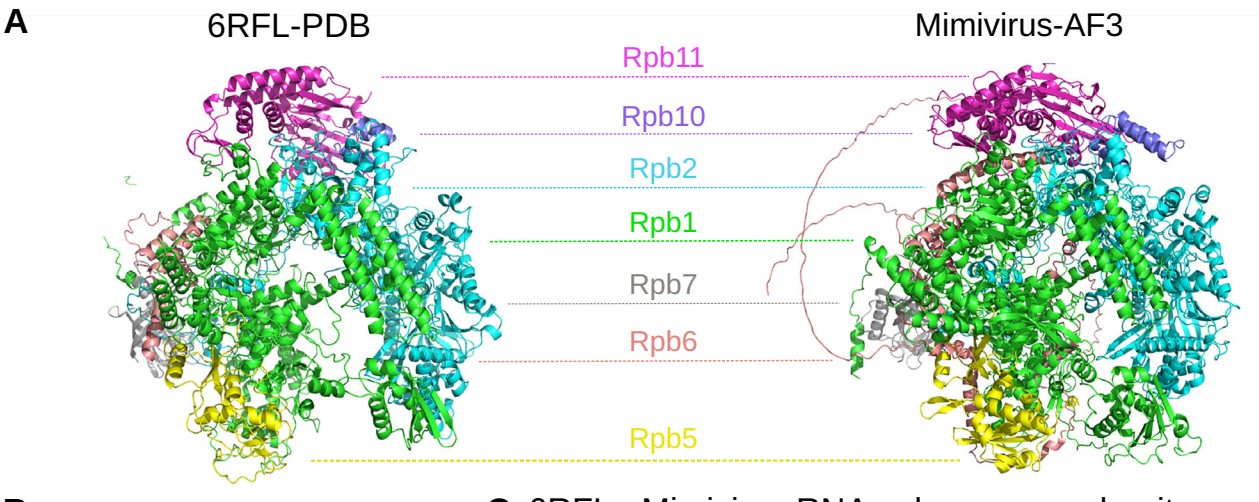

6RFL-PDB                                    Mimivirus-AF3

Rpb11
Rpb10
Rpb2
Rpb1
Rpb7
Rpb6

Rpb5

## B

R357b
6RFL_J

RMSD : 2.21A
over 48 residues

## C  6RFL - Mimivirus RNA polymerase subunits

| DNA Directed Polymerase subunit | 6RFL-PDB | Mimivirus | Blast E-value | Foldseek E-value |
|---|---|---|---|---|
| Rpb1 | 6RFL_A | R501 | 2.00E-46 | |
| Rpb2 | 6RFL_B | L244 | 3.00E-68 | |
| Rpb11 | 6RFL_C | R470 | - | 1.98E-07 |
| Rpb5 | 6RFL_E | L235 | - | 4.83E-06 |
| Rpb6 | 6RFL_F | R209 | - | 4.24E-02 |
| Rpb7 Nter | 6RFL_G | L376 | - | 1.37E-09 |
| Rpb10 | 6RFL_J | R357b | - | 1.23E-01 |

## D  R357b merge strategy

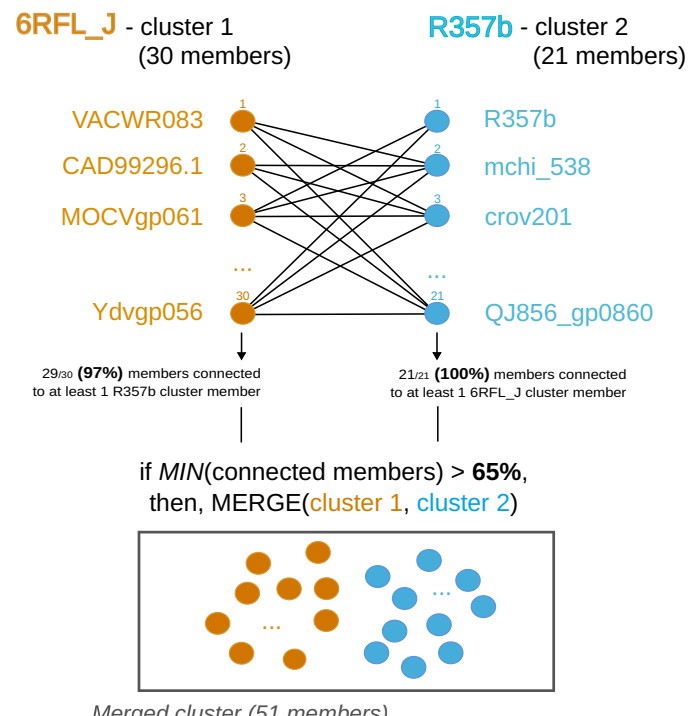

6RFL_J - cluster 1
(30 members)

R357b - cluster 2
(21 members)

| VACWR083 | R357b |
| CAD99296.1 | mchi_538 |
| MOCVgp061 | crov201 |
| ... | ... |
| Ydvgp056 | QJ856_gp0860 |

29/30 **(97%)** members connected
to at least 1 R357b cluster member

21/21 **(100%)** members connected
to at least 1 6RFL_J cluster member

if *MIN*(connected members) > **65%**,
then, MERGE(cluster 1, cluster 2)

*Merged cluster (51 members)*

## E  Merge step comparison

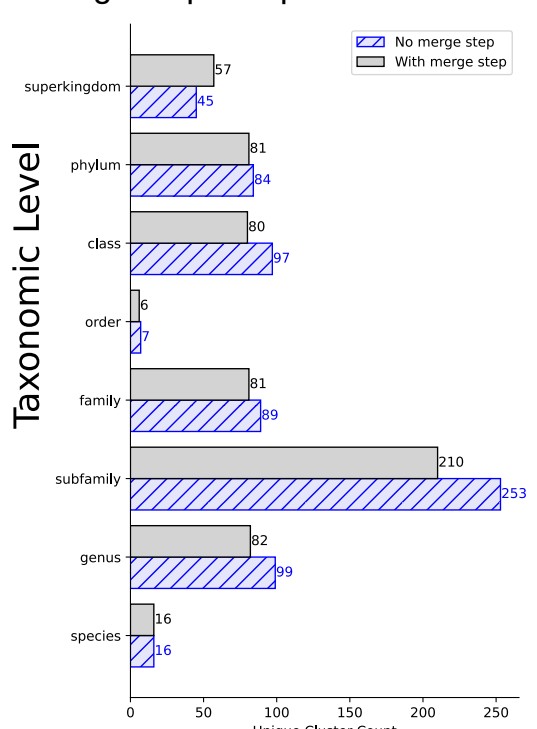

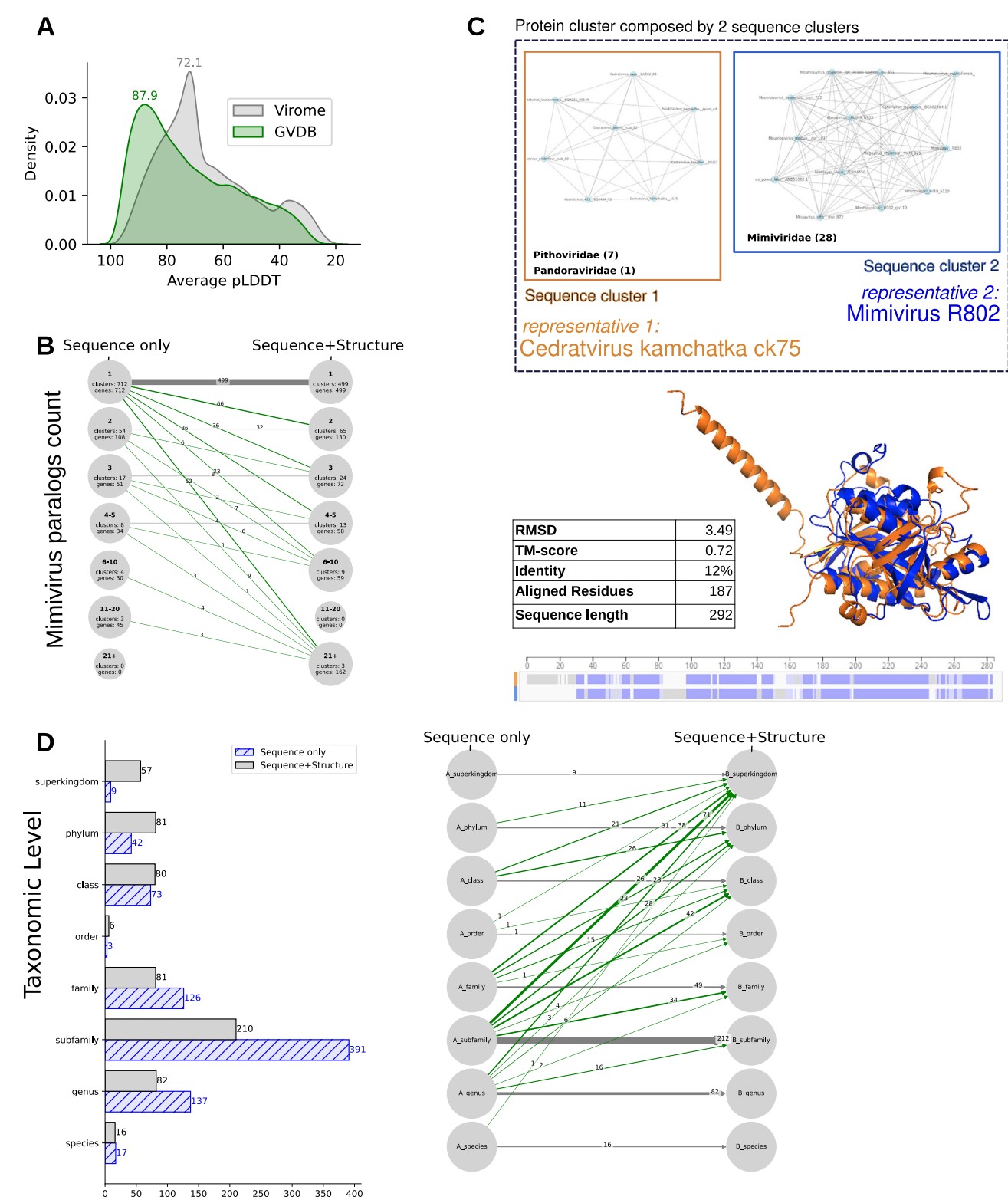

**Figure EV2.  Clustering enhancement combining sequence and 3D AF models.**

(**A**) Distribution of the pLDDT scores of AF models in the Virome and GVDB databases. (**B**) Distribution of the Number of paralogs in mimivirus, identified by Sequence-only or Sequence+Structure clustering strategy. (**C**) Example of 2 sequence clusters (*Pithoviridae/Pandoraviridae* and *Mimiviridae*) merged using sequence+structure clustering (TM-score 0.72 and 12% sequence identity). (**D**) LCA taxonomic levels for mimivirus protein clusters with Sequence-only or Sequence+structure clustering.

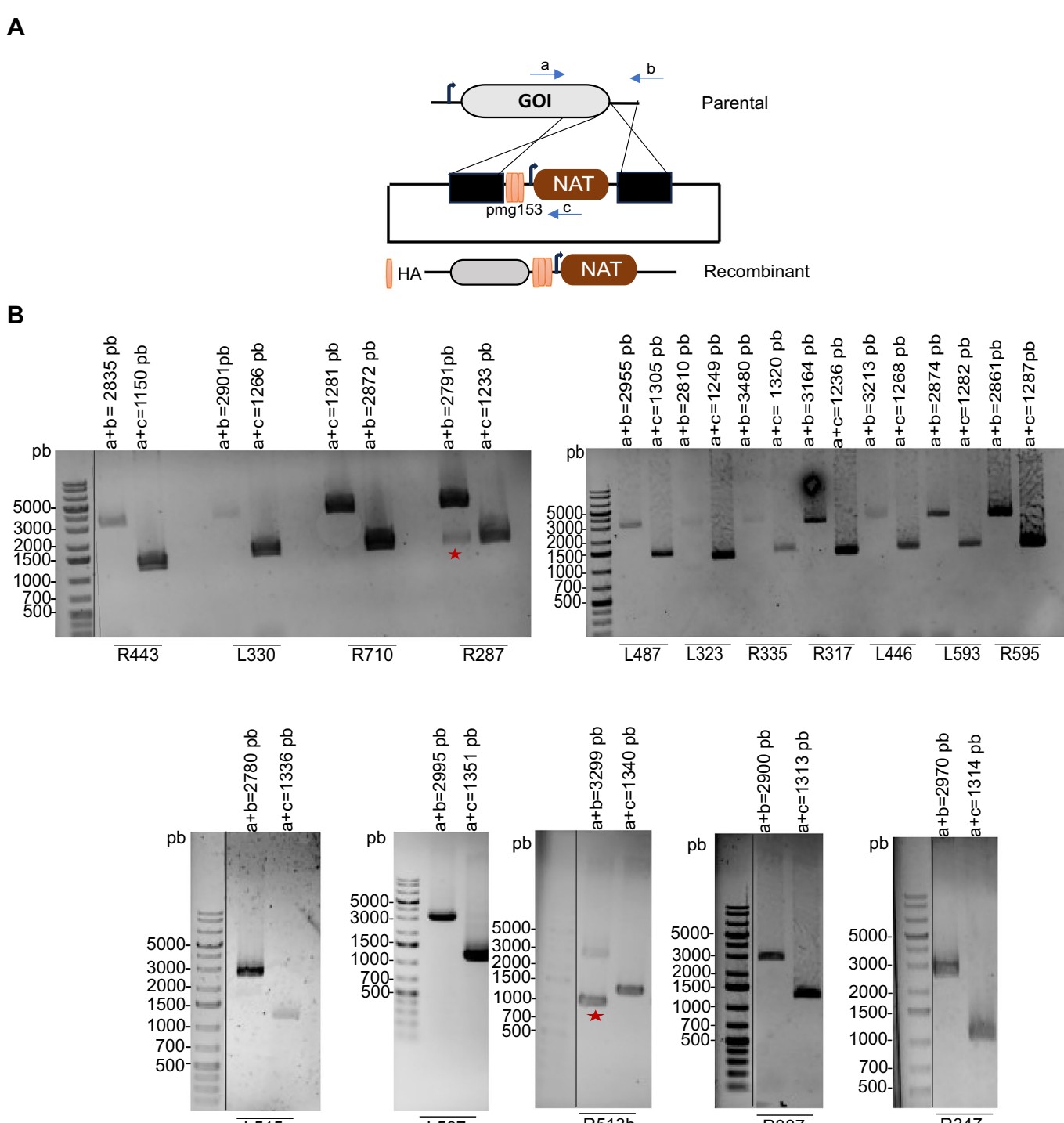

**Figure EV3. Generation of mimivirus recombinant viruses by homologous recombination.**

(A) Schematic representation of the vector and strategy used for endogenous HA tagging of mimivirus proteins. GOI: gene of interest. NAT: nourseothricin N-acetyl transferase selection cassette. Primer annealing locations are shown. (B) Confirmation of the clonality of mutants by PCR. Expected sizes are indicated in the figure: primers (a + b) used for genotyping on parental viruses and (a + c) for genotyping on recombinant viruses. Red stars highlight non-pure mutant clones.

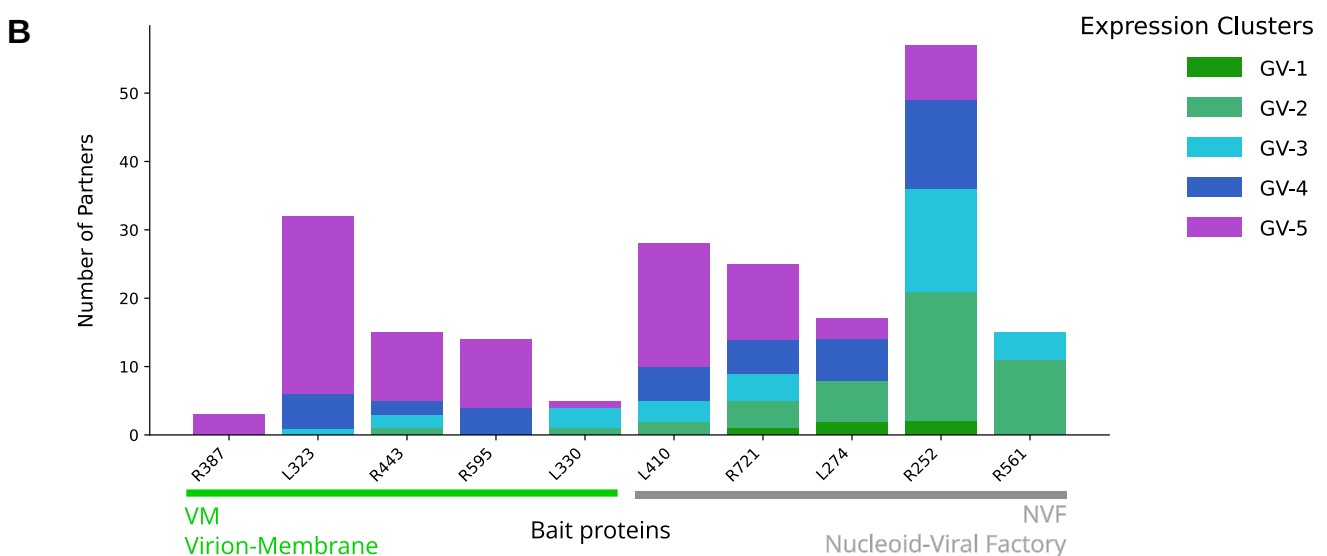

**Figure EV4.  Functional partitioning of the VM and NVF subnetworks mirroring temporal expression patterns.**

(A) Detailed network showing all connections of the IP-MS baits with their connected proteins. Expression cluster is shown as node colors from green (early genes, GV-1) to violet (late genes, GV-5). (B) Expression cluster counts for each IP-MS bait.

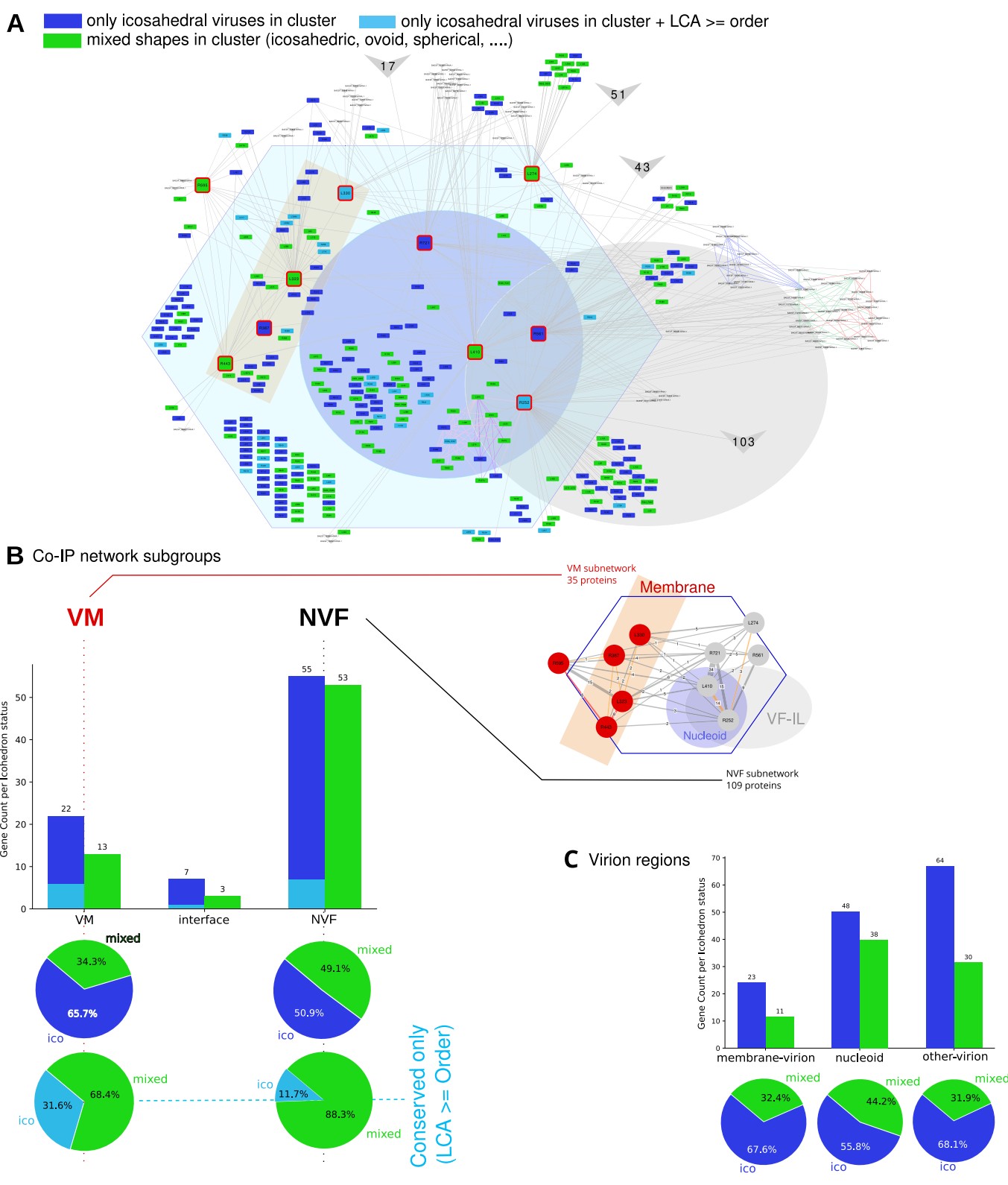

A

- only icosahedral viruses in cluster
- only icosahedral viruses in cluster + LCA >= order
- mixed shapes in cluster (icosahedric, ovoid, spherical, ....)

B Co-IP network subgroups

VM subnetwork
35 proteins

Membrane

NVF subnetwork
109 proteins

VM    NVF

Gene Count per Icohedron status

7    3

55    53

VM    interface    NVF

mixed

ico 65.7%    mixed 34.3%

ico 50.9%    mixed 49.1%

ico 31.6%    mixed 68.4%

ico 11.7%    mixed 88.3%

Conserved only (LCA >= Order)

C Virion regions

Gene Count per Icohedron status

23    11

48    38

64    30

membrane-virion    nucleoid    other-virion

ico 67.6%    mixed 32.4%

ico 55.8%    mixed 44.2%

ico 68.1%    mixed 31.9%

◀ **Figure EV5. Co-IP network analysis and virion shape.**

(A) Distribution of clusters with varying virion shapes as shown by node colors. Mimivirus proteins belonging to a cluster composed of icosahedral viruses are shown in blue. Light blue proteins indicate that the protein is at least conserved at the order level. Green nodes indicate proteins that belong to clusters enclosing mixed shapes viruses. (B) For each sub-network, clusters enclosing proteins from icosahedral-shaped viruses are in blue, and clusters enclosing mixed-shaped viruses are in green. The membrane (VM) sub-network shows a higher proportion of clusters from icosahedral-shaped viruses. (C) For each region, independently from the co-IP network, clusters with proteins from icosahedral-shaped viruses are in blue, and clusters enclosing mixed-shaped viruses are in green. The nucleoid region contains a higher fraction of mixed shape protein clusters than the membrane-virion region or the other-virion regions.

