## [Peer Review File · The EMBO Journal]

Elucidating the protein interaction network of one of the largest icosahedral capsids in the virosphere

Hela Safi, Alain Schmitt, Alwena Tollec, Lucid Belmudes, Agathe Colmant, Olivier Poirot, Sebastien Santini, Matthieu Legendre, Yohann Couté, Hugo Bisio, and Chantal Abergel

Corresponding authors: Chantal Abergel (Chantal.Abergel@igs.cnrs-mrs.fr) , Hugo Bisio (bisio@igs.cnrs-mrs.fr)

Review Timeline:

Submission Date:	10th Oct 25
Editorial Decision:	7th Dec 25
Revision Received:	21st Jan 26
Editorial Decision:	19th Feb 26
Revision Received:	4th Mar 26
Accepted:	20th Mar 26

Editor: Ieva Gailite

Transaction Report:

Dear Dr. Abergel,

Thank you for submitting your manuscript for consideration by the EMBO Journal. I apologise for the protracted assessment process for your manuscript due to delays in reviewer report submission. We have now received comments from a full set of reviewers, which are included below for your information.

As you will see, all reviewers are generally positive in their evaluation and appreciate the contribution of the study to the research field. They suggest only minor modifications, mainly aimed at the improvement of the textual and data presentation. Based on these positive assessments, I invite you to submit a revised manuscript in response to the comments by all reviewers.

When preparing your letter of response to the referees' comments, please bear in mind that this will form part of the Review Process File, and will therefore be available online to the community. For more details on our Transparent Editorial Process, please review our Editorial Policies: <https://link.springer.com/partners/embo-press/editorial-policies>

Please feel free to contact me if you have any further questions regarding the revision. Thank you for the opportunity to consider your work for publication. I look forward to your revision.

With best wishes,

leva

leva Gailite, PhD
Senior Scientific Editor
The EMBO Journal
Meyerhofstrasse 1
D-69117 Heidelberg
Tel: +4962218891309
i.gailite@embojournal.org

Read our guidance for manuscript revisions and related editorial policies: <https://link.springer.com/journal/44318/submission-guidelines#cms-Revised-submissions>

<https://media.springernature.com/original/springer-cms/rest/v1/content/27825798/data/v1>

- a point-by-point response to the referees' comments, with a detailed description of the changes made (as a word file).
- a word file of the manuscript text.
- individual production quality figure files (one file per figure)
- a complete author checklist
- Expanded View files (replacing Supplementary Information)
- a Reagents and Tools Table as part of the Methods section

We realize that it is difficult to revise to a specific deadline. In the interest of protecting the conceptual advance provided by the work, we recommend a revision within 3 months (7th Mar 2026). Please discuss the revision progress ahead of this time with the editor if you require more time to complete the revisions.

Referee #1:

In the study "Elucidating the interaction network of one of the largest icosahedral capsids in the virosphere" by Safi & Schmitt et al., the authors dive deep into the proteomic interactions of mimivirus virion morphogenesis and assembly, and provide a bounty of new insights for the whole giant virus (GV) scientific community. The introduction has a pleasant narrative flow and was enjoyable to read. The authors conduct in-depth structural predictions on the mimivirus proteome using AlphaFold (AF)-assisted analyses and clustering to infer a comprehensive mimivirus interactome. They accompany their in-silico analyses with well-documented experimental approaches, including endogenous tagging and co-immunoprecipitation with mass spectrometry. The analyses by Safi & Schmitt et al. not only reveal protein network conservation among mimiviruses, but also evolutionary conserved nodes and virion shape-specific adaptations with other icosahedral and non-icosahedral giant viruses. Next to the vast amount of new information, the authors also provide a webserver and clustering pipeline to allow others to browse for (giant) virus orthogroups, in a database with >46,500 added AF-predicted structures within the Nucleocytoviricota. Overall, this study is splendidly documented and displayed, and provides relevant and valuable new insights to the research field of Giant Viruses.

Major comments and questions

In case it is not already planned, I would consider moving certain sections of the material and methods to the supplementary information, for instance, the table with the reagents and tools.

Line 295-297: Has anyone ever performed an analysis to determine the selection pressure on mimivirus structural proteins, based on the dN/dS ratio for components like the DJR-MCP, for example? A full evolutionary analysis would probably burst the scope of this current paper. However, it might be something interesting to consider for future studies to elucidate whether a correlation exists between different mimivirus protein groups/clusters and their respective evolutionary rates.

Line 391: I assume the 0 hpi timepoint did not show any signal? Just to exclude potential autofluorescence in the 0.25 hpi sample.

Line 522: What modifications were made? If they are minor, they might as well be shortly described.

Figure 3: In Figure 3, I believe it would make more sense to rename 3D to 3C, and consequently make 3C to 3D, due to 3B's indicated connection with that bar plot. However, I wonder if the authors have considered making that large network into its own figure, since it already carries so much information on its own, and also, it is hard to read at the current size.

Minor comments

Generally:

- Write et al. in italics

Line 45: Swap Pandoraviridae and Pithoviridae in the sentence to match the order of the previous sentence

Line 53: add space between harbors and the

Line 535: Debris WAS removed

Line 575: Instead: "Precursor and fragment mass error tolerances were set at 10 and 20 ppm, respectively."

Line 578: Something went wrong with the formatting of the Bouyssié et al, 2020 reference here

Line 584: Same as in Line 578 for Wieczorek et al, 2017

Line 591: Limma should be cited

Line 619: ..., not used in the original Nomburg study, (add another comma or put in between em dashes)

Line 630: Wrong reference formatting (Legendre et al, 2011)

Line 634: ... taxonomic level (instead of taxonomical)

Line 674: Cytoscape in capital, and reference is missing

Line 683: Cytoscape in capital

Line 685-687: Use quotation marks to refer to the mentioned sections, e.g. (see "Virion, nucleoid and VF protein sets", as well as "Prediction of transmembrane domain containing proteins sections").#

Line 688: The host nodes (host in lower case)

Line 691: castellanii in lowercase

Line 725: Proteome is missing an "e" at the end

Referee #2:

This manuscript investigates the protein interaction networks of Mimivirus, a model organism for giant viruses, which remain among the most enigmatic entities in the virosphere. The authors generated extensive experimental data on protein-protein interactions during viral infection and integrated these data with an extended and comprehensive protein orthologue database based on sequence and (predicted) structural similarities. This approach enables the identification of conserved protein families and interaction modules that may drive viral processes within the infected host cell.

This is a comprehensive cutting-edge study that is innovative both experimentally and conceptually. It provides unprecedented insights into protein-protein interactions, uncovers key interaction hubs, and highlights protein sets likely functioning together in the virocell during Mimivirus infection.

This work represents a significant advancement for the field of virology, particularly in understanding the biology and evolution of giant viruses.

I have no major comments but would like to note that the research field would benefit from an easier access to the data generated in the study (see specific comment below).

1. L 78, 82: Define AF, IP-MS when used first! Double check this for all abbreviations!
2. References to EV figures (or the naming in the uploaded files) seem to be partially incorrect, e.g. L 134 reference to EV2AB; L 139 reference to EV1D. Please double check all EV figure references!
3. L 140/Figure 1E: Different statements regarding the highest degree of conservation, which is corresponding to 21% of the predicted proteome in the main text and to 22% in the figure legend. Where is this represented in the figure?
4. Figure 1 title should read "The predicted structural proteome ..."
5. Figure 1E: The title of this panel should reflect that these plots are about Mimivirus proteins. The sublevel composition matrices are not very informative, can be omitted to focus the figure on the main findings and increase clarity. The legend should mention how presence/absence of proteins in the virion was determined.
6. Western blot analysis: Some of the Western blots are not very clear; can anyways be moved to the supplementary material to reduce complexity of Figure 2. Mention the IF control experiment with mock-infected cells in the legend.
7. GO term enrichment analysis: Why is Figure 3C referenced here? In contrast to the statement in the main text (L 250), no data is available for R561 in Table EV3. In the figure legend, state that subnetworks inferred from by homology with PDB structures are indicated by colored edges.
8. Complementation of the R443 KO mutant with the WT (HA tagged) gene showed a similar fitness loss in the competition assay as the KO mutant. How can this be explained?
9. L410-derived IF signals are hardly visible in Figure 6. Please either provide images with higher magnifications, or move to supplement, and restrict the Figure to the plots showing the signal quantification; panels B and C seem to be largely redundant, suggest to omit B. Why was the 5h time point not quantified?
10. The GV-clusters webserver is a laudable idea, but in its present form the webserver is only partially functional and helpful: The search by protein identifier would need to be a free text search form to being able to find anything. The sequence similarity search generates results but my test search for a highly conserved protein returns a single hit, and the download function didn't work. I encourage the authors to make the entire data set available, e.g. as a Foldseek database, and perhaps follow the example of Nomburg et al. (2024) for providing search and browse functionality.

Referee #4:

In this manuscript, authors integrated bioinformatics, genetics, and interactomics to elucidate the protein functions and networks that comprise Mimivirus (the world's largest icosahedral virus). To this end, they clustered proteins using the AlphaFold model and mapped the protein-protein interaction network using co-immunoprecipitation mass spectrometry. These results have brought about new advances in elucidating the morphology and evolutionary relationships of large viruses. Although current

results are limited to a few viral proteins, this pioneering research will expand future proteomics research into a wide range of fields in life sciences and human pathology. For these reasons, we believe this manuscript is worthy of publication in this journal and recommend its publication.

Minor revisions are as follows:

- 1) Line 53: harborsthe -> harbors the
- 2) Line 251: Abbreviations of BP, CC, MF should be cleared.
- 3) 354: Include ".".
- 4) 395: Remove one of two "at".

Point-by-point reply: EMBOJ-2025-122698

Elucidating the protein interaction network of one of the largest icosahedral capsids in the virosphere (Hela Safi, Alain Schmitt, Alwena Tollec, Lucid Belmudes, Agathe M. G. Colmant, Olivier Poirot, Sebastien Santini, Matthieu Legendre, Yohann Couté, Hugo Bisio, Chantal Abergel).

Referee #1:

In the study "Elucidating the interaction network of one of the largest icosahedral capsids in the virosphere" by Safi & Schmitt et al., the authors dive deep into the proteomic interactions of mimivirus virion morphogenesis and assembly, and provide a bounty of new insights for the whole giant virus (GV) scientific community. The introduction has a pleasant narrative flow and was enjoyable to read. The authors conduct in-depth structural predictions on the mimivirus proteome using AlphaFold (AF)-assisted analyses and clustering to infer a comprehensive mimivirus interactome. They accompany their in-silico analyses with well-documented experimental approaches, including endogenous tagging and co-immunoprecipitation with mass spectrometry. The analyses by Safi & Schmitt et al. not only reveal protein network conservation among mimiviruses, but also evolutionary conserved nodes and virion shape-specific adaptations with other icosahedral and non-icosahedral giant viruses. Next to the vast amount of new information, the authors also provide a webserver and clustering pipeline to allow others to browse for (giant) virus orthogroups, in a database with >46,500 added AF-predicted structures within the Nucleocytoviricota. Overall, this study is splendidly documented and displayed, and provides relevant and valuable new insights to the research field of Giant Viruses.

Major comments and questions

In case it is not already planned, I would consider moving certain sections of the material and methods to the supplementary information, for instance, the table with the reagents and tools.

We thank the reviewer for this suggestion. However, EMBO Press requires that the Reagents and Tools Table remains within the main Methods section, as stated in the editorial instructions. We have therefore kept the table in the manuscript accordingly.

A full evolutionary analysis would probably burst the scope of this current paper. However, it might be something interesting to consider for future studies to elucidate whether a correlation exists between different mimivirus protein groups/clusters and their respective evolutionary rates.

We agree that investigating the evolutionary rates of the different mimivirus protein groups would be very interesting, but this analysis goes beyond the scope of the present study. We appreciate the suggestion and will keep this perspective in mind for future work.

Line 391: I assume the 0 hpi timepoint did not show any signal? Just to exclude potential autofluorescence in the 0.25 hpi sample.

As stated in the main text (manuscript .docx lines 387-389, manuscript .pdf lines 393-395), no signal was detected at 0 hpi. At this stage, the viral core remains enclosed within intact capsids and the L410 protein is therefore inaccessible to antibody detection. This also confirms that the signal detected at 0.25 hpi does not result from autofluorescence.

Line 522: What modifications were made? If they are minor, they might as well be shortly described.

The method implemented in this study is fully described in the manuscript. The reference is cited to acknowledge the original protocol, but all steps, including the minor adaptations, are explicitly detailed in the Methods section (manuscript .docx lines 519-535, manuscript .pdf lines 528-544).

Figure 3: In Figure 3, I believe it would make more sense to rename 3D to 3C, and consequently make 3C to 3D, due to 3B's indicated connection with that bar plot. However, I wonder if the authors have considered making that large network into its own figure, since it already carries so much information on its own, and also, it is hard to read at the current size.

Originally, we chose to have 3D following 3C (large network with localization information such as Virion, Core, ...) because 3D is a synthesis of 3C with respect to the localization information. As notified by the reviewer, 3D is also related to 3B as per the sub-networks definition (VM or CVF). Finally, we are now following the reviewer's suggestion for readability, and swapped 3D and 3C. We agree with the reviewer that this detailed network figure carries a lot of information, which is why we provided the Cytoscape files ready to be viewed (downloadable from Zenodo), to access all information interactively.

Minor comments

Generally:

- Write et al. in italics

We have formatted all references according to EMBO Press style guidelines.

Line 45: Swap Pandoraviridae and Pithoviridae in the sentence to match the order of the previous sentence

This has now been corrected

Line 53: add space between harbors and the

This has now been corrected

Line 535: Debris WAS removed

This has now been corrected

Line 575: Instead: "Precursor and fragment mass error tolerances were set at 10 and 20 ppm, respectively."

This has now been corrected

Line 578: Something went wrong with the formatting of the Bouyssi² et al, 2020 reference here

This has now been corrected

Line 584: Same as in Line 578 for Wieczorek et al, 2017

This has now been corrected

Line 591: Limma should be cited

The citation is now included

Line 619: ..., not used in the original Nomburg study, (add another comma or put in between em dashes)

This has now been corrected

Line 630: Wrong reference formatting (Legendre et al, 2011)

This has now been corrected

Line 634: ... taxonomic level (instead of taxonomical)

This has now been corrected

Line 674: Cytoscape in capital, and reference is missing

This has now been corrected

Line 683: Cytoscape in capital

This has now been corrected

Line 685-687: Use quotation marks to refer to the mentioned sections, e.g. (see "Virion, nucleoid and VF protein sets", as well as "Prediction of transmembrane domain containing proteins sections").#

This has now been corrected

Line 688: The host nodes (host in lower case)

This has now been corrected

Line 691: castellanii in lowercase

This has now been corrected

Line 725: Proteome is missing an "e" at the end

This has now been corrected

Line 942: Wrong reference formatting (Nomburg et al, 2024)

This has now been corrected in the caption of figure S1.

Referee #2:

This manuscript investigates the protein interaction networks of Mimivirus, a model organism for giant viruses, which remain among the most enigmatic entities in the virosphere. The authors generated extensive experimental data on protein-protein interactions during viral infection and integrated these data with an extended and

comprehensive protein orthologue database based on sequence and (predicted) structural similarities. This approach enables the identification of conserved protein families and interaction modules that may drive viral processes within the infected host cell.

This is a comprehensive cutting-edge study that is innovative both experimentally and conceptually. It provides unprecedented insights into protein-protein interactions, uncovers key interaction hubs, and highlights protein sets likely functioning together in the virocell during Mimivirus infection.

This work represents a significant advancement for the field of virology, particularly in understanding the biology and evolution of giant viruses.

I have no major comments but would like to note that the research field would benefit from an easier access to the data generated in the study (see specific comment below).

1. L 78, 82: Define AF, IP-MS when used first! Double check this for all abbreviations!

This has now been corrected for AF and IP-MS, and all abbreviations have been checked.

2. References to EV figures (or the naming in the uploaded files) seem to be partially incorrect, e.g. L 134 reference to EV2AB; L 139 reference to EV1D. Please double check all EV figure references!

These specific Figures references have been corrected, and we have carefully checked all other EV references too.

3. L 140/Figure 1E: Different statements regarding the highest degree of conservation, which is corresponding to 21% of the predicted proteome in the main text and to 22% in the figure legend. Where is this represented in the figure?

We thank the reviewer for pointing out that inconsistency. We corrected the fractions both in the text (manuscript .docx line 140, manuscript .pdf line 143) and the figure caption (manuscript .docx line 821, manuscript .pdf line 836), indicating the raw counts used to estimate the fraction of sub-family protein clusters (210/613=34 % as observed in fig. 1E)

4. Figure 1 title should read "The predicted structural proteome ..."

This has now been corrected

5. Figure 1E: The title of this panel should reflect that these plots are about Mimivirus proteins. The sublevel composition matrices are not very informative, can be omitted to focus the figure on the main findings and increase clarity. The legend should mention how presence/absence of proteins in the virion was determined.

We thank the reviewer for that suggestion to improve the clarity of this Figure. We have changed the title of this panel in Fig. 1E to «Mimivirus protein clusters conservation », and

moved the LCA (Last Common Ancestor) mention to the y-axis label. To keep the focus on the main findings of Figure 1E, we visually separated the sublevel composition matrices with a light blue background color to add clarity.

The presence/absence of proteins from the virion was determined by using proteomics data from Villalta et al. as described in the Methods section. This mention was added in the figure caption (manuscript .docx lines 820-821, manuscript .pdf lines 834-836) as suggested by this reviewer.

6. Western blot analysis: Some of the Western blots are not very clear; can anyways be moved to the supplementary material to reduce complexity of Figure 2. Mention the IF control experiment with mock-infected cells in the legend.

This has now been corrected. Western blot data are now moved to Figure S2 (Legend in red to highlight the addition).

7. GO term enrichment analysis: Why is Figure 3C referenced here? In contrast to the statement in the main text (L 250), no data is available for R561 in Table EV3.

In the figure legend, state that subnetworks inferred from by homology with PDB structures are indicated by colored edges.

We initially referenced Figure 3C because it showed the functional categories of all partners of the bait proteins that were subjected to Go terms enrichment analysis. We removed that reference as suggested by the reviewer, for clarity.

We would like to point out that the Table EV3 contains 2 tabs. One for the viral proteins Go enrichment analysis, and the other for the host proteins. While the « virus » tab does not contain R561 enrichment results, the « host » tab does, as stated in the main text. We now refer to the 2 Tabs when calling this table (manuscript .docx line 244, manuscript .pdf line 247).

As suggested, we added a mention in the figure legend, to state that PDB complexes obtained by homology are indicated by colored edges.

8. Complementation of the R443 KO mutant with the WT (HA tagged) gene showed a similar fitness loss in the competition assay as the KO mutant. How can this be explained?

In the competition assays, fitness is measured as a relative ratio between two competing viruses. We compared R443 KO versus WT, and R443 KO versus the cis-complemented strains expressing R443 (WT (CXXC), CXXS and SXXC). The similar fitness loss observed in all cases indicates that the complemented virus behaves like the WT virus in this assay, consistent with R443 being non-essential for viral replication but providing fitness to the virus. A sentence has been added to this section (manuscript .docx lines 367-369, manuscript .pdf lines 372-374).

9. L410-derived IF signals are hardly visible in Figure 6. Please either provide images with higher magnifications, or move to supplement, and restrict the Figure to the plots showing

the signal quantification; panels B and C seem to be largely redundant, suggest to omit B. Why was the 5h time point not quantified?

This has now been corrected. The immunofluorescence images have been moved to the Expanded View section (now shown in Fig. S7), and the main figure has been simplified by retaining only panel C. High-magnification insets were added in Fig. S7 to better visualize the punctate L410 signals (legend in red to highlight the addition). The 5 hpi time point was not quantified because, at this stage, de novo synthesis of L410 has started and the protein becomes abundantly redistributed in viral factories, resulting in numerous puncta (manuscript .docx lines 396 to 402, manuscript .pdf lines 403-409).

10. The GV-clusters webserver is a laudable idea, but in its present form the webserver is only partially functional and helpful: The search by protein identifier would need to be a free text search form to being able to find anything. The sequence similarity search generates results but my test search for a highly conserved protein returns a single hit, and the download function didn't work.

I encourage the authors to make the entire data set available, e.g. as a Foldseek database, and perhaps follow the example of Nomburg et al. (2024) for providing search and browse functionality.

We thank the reviewer for his/her feedback on the web server that helped us improve the user interface. We have modified the Search field with autocompletion to enable free text searches for every protein in the Virome+GV database. The user input is then used to filter protein Ids and also functional annotations that were available in public databases (eg. NCBI).

We point out that the similarity search returns a list of protein clusters containing members with sequence similarity to the query. If all search results are in the same cluster, there is only one hit displayed, even if there are numerous protein hits. For clarity, we have added a brief user help description on the web server's main page.

The clusters' PDB members download feature was working for most clusters, but with a size limitation (server response too high) that we corrected thanks to the reviewers' feedback.

The availability of the entire dataset, for example, as Foldseek database is already done via the zenodo uploads (see Data availability section: (<https://doi.org/10.5281/zenodo.17296301>))

Referee #4:

In this manuscript, authors integrated bioinformatics, genetics, and interactomics to elucidate the protein functions and networks that comprise Mimivirus (the world's largest icosahedral virus). To this end, they clustered proteins using the AlphaFold model and

mapped the protein-protein interaction network using co-immunoprecipitation mass spectrometry. These results have brought about new advances in elucidating the morphology and evolutionary relationships of large viruses. Although current results are limited to a few viral proteins, this pioneering research will expand future proteomics research into a wide range of fields in life sciences and human pathology. For these reasons, we believe this manuscript is worthy of publication in this journal and recommend its publication.

Minor revisions are as follows:

1) Line 53: harborsthe -> harbors the

This has now been corrected

2) Line 251: Abbreviations of BP, CC, MF should be cleared.

This has now been corrected

3) 354: Include ".".

This has now been corrected

4) 395: Remove one of two "at".

This has now been corrected

Dear Dr. Abergel,

Thank you for submitting a revised version of your manuscript. I apologise for the extended evaluation process - I have attempted to contact reviewer #2 for re-assessment, but they have unfortunately been unresponsive. Since all reviewers have marked the requested changes as minor, I have now gone through the incorporated revisions and find them generally reasonable.

There now remain only a few editorial and formatting points that need to be addressed before I can extend official acceptance of the manuscript:

1. Please note that it is The EMBO Journal policy for the transcript of the editorial process (containing referee reports and your response letters) to be published as an online supplement to each paper. If you should prefer removal of any referee-only figures included in the point-by-point response(s), e.g. because they may still be used for future publication or because they have been reproduced from published work by others, please do let us know immediately via response email.

More information is available here: https://www.embopress.org/transparent-process#Review_Process

2. Please make sure that the order of the sections in the manuscript is as follows: Title page - Abstract & Keywords - Introduction - Results - Discussion - Methods - Data Availability - Acknowledgments - Disclosure and Competing Interests Statement - References - Figure Legends - (Main Tables with legends if applicable) - Expanded View Figure Legends.

1. Please check that the funding information is correct and identical both in the manuscript and our online system. Currently, full information appears missing in our online system for the following grants: grant agreement No 832601, ERC-2018-ADG and grant agreement No. 101160452, ERC-2024-STG.

3. CRediT has replaced the traditional author contributions section because it offers a systematic, machine-readable author contributions format that allows for more effective research assessment. Please remove the Authors Contributions from the manuscript and use the free text boxes beneath each contributing author's name in our online submission system to add specific details on the author's contribution. More information is available in our guide to authors.

4. In the Data Availability section, please add a resolvable link to the proteomics dataset. Please also include code availability in this section. More information about the format of this section can be found here: <https://link.springer.com/partners/embopress/editorial-policies#Data%20availability%20statement>.

5. Please turn Tables EV3 and EV5 into Datasets and update the nomenclature accordingly throughout the manuscript.

6. Please remove EV table and dataset legends from the manuscript; their legends should be included in the corresponding EV table/dataset file itself. In the case of datasets, legends should be included in a separate sheet. The legend for Dataset EV3 is currently missing in the file.

7. For Table EV4, the title in the table file is mislabelled as "Table EV1", please correct.

8. Please provide Appendix as a pdf file. In the title page, please include the title "Appendix for 'Ms title'". Please remove the page labelled 'Appendix Supplementary Figures' and remove the tracked changes/highlights.

9. Please update the callouts for Appendix figures to Appendix Figure S1, etc. throughout the manuscript.

11. In our standard image integrity check, we noticed image panel reuse for the following figure panels:

- Figure 2 and 5C, lane R443.

- Appendix Figure S2 - control lanes appear reused.

If this was intentional, please indicate the reuse in the figure legends.

12. In our standard image check, we noticed that the resolution of the submitted figures is low for blot and microscopy images. This reduction in resolution is commonly caused by converting original 16-bit TIFF files to RGB format for publication. While this is not inherently problematic, it can raise concerns about image integrity for critical readers.

To avoid any misunderstanding and to meet EMBO Press standards, we kindly ask that you:

* Resubmit the complete figure set at the captured original data resolution.

* Apply the same resolution standards to the blot source data files, which are also currently below the required quality threshold.

Please upload the blot source data files as .tiff or pdf. Please do not use PowerPoint.

13. Papers published in The EMBO Journal are accompanied online by a 'Synopsis' to enhance discoverability of the manuscript. It consists of A) a short (1-2 sentences) summary of the findings and their significance, B) 3-4 bullet points highlighting key results and C) a synopsis image that is 550x300-600 pixels large (width x height, jpeg or png format). You can either show a model or key data in the synopsis image. Please note that the image size is rather small and that text needs to be readable at the final size.

With best wishes,

leva

leva Gailite, PhD
Senior Scientific Editor
The EMBO Journal
Meyerhofstrasse 1
D-69117 Heidelberg
Tel: +4962218891309
i.gailite@embojournal.org

Read our guidance for manuscript revisions and related editorial policies: <https://link.springer.com/journal/44318/submission-guidelines#cms-Revised-submissions>

We realize that it is difficult to revise to a specific deadline. In the interest of protecting the conceptual advance provided by the work, we recommend a revision within 3 months (20th May 2026). Please discuss the revision progress ahead of this time with the editor if you require more time to complete the revisions.

The authors addressed the remaining editorial issues.

Dear Dr. Abergel,

Thank you for incorporating the final editorial and formatting requests in the manuscript. I am now pleased to inform you that your manuscript has been accepted for publication. Congratulations with a nice study!

Before we forward your manuscript to our publishers, we would like to propose some edits in the manuscript title, abstract and synopsis. I have also written a short blurb that will accompany the title of your manuscript in our online system. Please take a look at the proposed text changes in the attached text file and let me know if any corrections are needed.

You may qualify for financial assistance for your publication charges - either via a Springer Nature fully open access agreement or an EMBO initiative. Check your eligibility: <https://link.springer.com/journal/44318/how-to-publish-with-us>

If you have any questions, please do not hesitate to contact the Editorial Office or me directly. Thank you for this interesting contribution to The EMBO Journal!

Best wishes,

Ieva

Ieva Gailite, PhD
Senior Scientific Editor
The EMBO Journal
Meyerhofstrasse 1
D-69117 Heidelberg
Germany
i.gailite@embojournal.org